

# Impact of the wind field at the complex terrain site Perdigão on the surface pressure fluctuations of a wind turbine

Florian Wenz[1], Judith Langner[2], Thorsten Lutz[1], and Ewald Krämer[1]

[1]University of Stuttgart, Institute of Aerodynamics and Gas Dynamics (IAG), Pfaffenwaldring 21, 70569 Stuttgart, Germany
[2]WRD Management Support GmbH, Customer Support, Teerhof 59, 28199 Bremen, Germany

**Correspondence:** Florian Wenz (florian.wenz@iag.uni-stuttgart.de)

**Abstract.** The influence of turbulent inflow, as it occurs in complex terrain, on the unsteady surface pressure distributions on a wind turbine is investigated numerically. A method is presented that enables an accurate reproduction of the inflow to the turbine in the complex terrain in Perdigão, Portugal. For this purpose, a precursor simulation with the steady-state atmospheric computational fluid dynamics (CFD) code *E-Wind* and a high-resolution Delayed Detached Eddy Simulation (DDES) with
*FLOWer* is performed. The conservation of the flow field is validated by a comparison with measurements from the 2017 field campaign in Perdigão. Then, the resolved fluid-structure coupled generic wind turbine I82 is included in the *FLOWer* simulation to investigate the impact of the complex terrain inflow on the surface pressure fluctuations on tower and blades. A comparison with simulations of the same turbine in flat terrain with simpler inflows shows that the turbine in complex terrain has a significantly different vortex shedding at the tower, which dominates the periodic pressure fluctuations at the tower sides
and back. However, the dominant source of periodic pressure fluctuations on the upper part of the tower, the blade-tower interaction, is hardly altered by the terrain flow. The pressure fluctuations on the blade have a rather broadband characteristic, caused by the interaction of the leading edge with the inflow turbulence. In general, it is shown that a sophisticated DDES of the complex terrain plays a decisive role in the unsteady aerodynamics of the turbine, due to its specific flow characteristic.

## 1 Introduction

In the course of the onshore expansion of wind energy, more and more wind turbines are erected in complex terrain. The disturbances of the inflow angle, the strong turbulences and the inhomogeneity of the wind field that occur there pose a challenge for the prediction of turbine loads, performance and noise emission. Especially the low-frequency acoustic emissions are controversially discussed in the context of public acceptance of wind turbines in onshore wind parks. The basis for an accurate prediction of acoustic low-frequency emissions is the correct simulation and understanding of their aerodynamic
source, namely the surface pressure fluctuations on the tower and blades (Yauwenas, 2017; Klein, 2019). These are caused by the blade-tower interaction, the inflow turbulence and the vortex shedding, all of which are affected by the surrounding terrain and its specific flow field. The increase in computational resources enables high-fidelity simulations to capture more and more of these aerodynamic interactions in a complex terrain site and to evaluate the corresponding phenomena.



## 1.1 Numerical approaches for complex terrain and wind turbine simulations

Reliable methods for predicting flow characteristics are of great importance for profound site assessment, especially in complex terrain. Flow over hills is accelerated, can cause recirculation regions and turbulence characteristics are altered, all of which have been studied in research for decades, as the overview of Belcher and Hunt (1998) shows. These effects hold positive potential in terms of wind turbine performance, but also bear risks. In industry, computationally cheap approaches, such as steady-state Reynolds-averaged Navier–Stokes (RANS) simulations (Alletto et al., 2018), are needed for assessing risky turbine

position in complex terrain. Large-scale meteorological effects are often captured by Large Eddy Simulations (LES) with meteorological codes such as the Weather Research and Forecasting (WRF) model, which only coarsely resolve site-specific terrain features (Wagner et al., 2019). To capture the unsteady effects occurring in the atmospheric boundary layer and their interaction with complex terrain, unsteady RANS (URANS) simulations, as in Koblitz (2013), can be used. If the focus is on resolving the boundary layer or on the aerodynamic interaction of the inflow with the turbine, hybrid RANS/LES models are

necessary, since the small-scale vortices must be resolved. Bechmann and Sørensen (2010) simulated the flow over a hill with a hybrid formulation similar to the Detached Eddy Simulation (DES) with good results especially for the turbulence level. Schulz et al. (2016) conducted Delayed Detached Eddy Simulations (DDES) to evaluated the effect of complex terrain on the performance of a wind turbine, and the general suitability of DDES for detailed investigations of wind turbine aerodynamics is demonstrated by Weihing et al. (2018). Sørensen and Schreck (2014) performed DDES and URANS simulations of the NREL

Phase-VI rotor and found that although DDES does not improve the quality of the mean power prediction, it significantly increases the accuracy of the predicted load spectra compared to URANS. For the overall objective of investigating low-frequency acoustic emissions, which are strongly dependent on unsteady loads, it is therefore highly advisable to use DDES.

## 1.2 The complex terrain site Perdigão

A widely studied complex terrain site in the field of wind energy is the double ridge in Perdigão in central Portugal. The site

consists of two parallel, well-exposed ridges, each overlooking the surrounding area by about $300\,\mathrm{m}$. A single wind turbine has been erected on the southwestern ridge. A detailed description of the orography and vegetation at the site can be found in Vasiljević et al. (2017).

During the 2017 field campaign in Perdigão, a comprehensive set of measurement data of the flow over the complex terrain as well as of the behaviour of the wind turbine within it was collected. The measurement equipment used ranges from met

masts to lidars and microphones (Fernando et al., 2019). This campaign was part of the New European Wind Atlas (NEWA) funded by the European Union, which provides maps of wind statistics in complex terrains that can be used as a benchmark for site assessment. The NEWA experiments (Mann et al., 2017) consist of five measurement campaigns at complex sites, of which Perdigão was the most extensive, and provide, among other things, detailed microscale validation data for simulations. In order to obtain accurate simulation results, the terrain model on which the computational fluid dynamics (CFD) simulations

are based must be correspondingly detailed. Therefore, Palma et al. (2020) created a detailed digital terrain model (DTM) with a resolution of $2\,\mathrm{m}$, which includes orography and vegetation.





In many studies, simulations of the flow field in Perdigão have already been carried out to investigate the effects of orography, vegetation, thermal stratification as well as meteorological effects in general. Wagner et al. (2019) performed nested LES with WRF for the Iberian Peninsula with a highest resolution of $200\,\mathrm{m}$ around Perdigão covering almost 50 days. They showed that frequent nocturnal low-level jets over the double ridge from northeast already develop in Spain and that the southwest wind during the day experiences a clockwise wind turning. Coupled WRF and URANS simulations were used by Olsen (2018) to include changing weather patterns as well as local orograhic and surface effects. Characteristic eddy-structures behind the ridges were observed and with a finest mesh resolution of $80\,\mathrm{m}$ the mean wind speed was captured well. Steady-state RANS calculations were used by Palma et al. (2020) to discuss the impact of the resolution of the terrain model as well as of the CFD mesh on the local flow. They found that the flow in the valley was most affected by the resolution and recommend a resolution below $40\,\mathrm{m}$. Salim Dar et al. (2019) performed LES of the double ridge in Perdigão with a resolution of $10\,\mathrm{m}$ including the wind turbine, being represented with an actuator-disc model. They investigated the wake behaviour and found that the shape of the velocity deficit profile is preserved in downstream direction even in complex terrain, which is known as self-similarity. In addition, they found that the streamwise velocity at hub height varies with the change in terrain characteristics caused by a change in resolution. More detailed simulations of the interaction between turbulent terrain flow and local wind turbine aerodynamics using a fully resolved turbine in Perdigão have not been published to the authors' knowledge.

### 1.3   Scope and objectives

The influence of terrain flow on the unsteady pressure distributions on the turbine surface is investigated in order to examine the mechanisms of low-frequency noise generation in complex terrain using the example of the double ridge in Perdigão. A method is presented that aims to enable an accurate reproduction of the inflow to the turbine in complex terrain. For this purpose, an interface is created from the atmospheric CFD code *E-Wind* to the high-resolution DDES with *FLOWer* in order to provide an unsteady wind field at the domain inlet for it. The conservation of the flow field is evaluated by means of mean velocities and turbulence statistics and validated by a comparison with measurements in Perdigão. Furthermore, the necessary properties of a numerical setup including vegetation for a numerically stable and high-quality DDES of the complex terrain are given.

Then, the fluid-structure coupled generic turbine I82 (Arnold et al., 2020) with aero-servo-elastic similarity to the actual turbine at the site is included in the *FLOWer* simulation. The resulting aerodynamic effects, in particular the unsteady pressure distributions on tower and blade, are investigated. The observations are compared with the results of DDES of the same turbine in flat terrain with uniform inflow as well as a turbulent inflow generated from *E-Wind* results at the turbine position. In this way it can be assessed whether the sophisticated DDES of the complex terrain plays a decisive role in the unsteady aerodynamics of the turbine.

### 2   Numerical tools

The process chain for the calculation of unsteady aerodynamics under site- and situation-specific inflow in complex terrain comprises several solvers. The atmospheric steady-state CFD RANS solver *E-Wind* for the simulation of the Perdigão site





provides the inflow conditions for unsteady high-resolution DDES of the turbine near-field with the CFD solver *FLOWer*. The geometrically resolved turbine can be included in this simulation and a time-accurate coupling to the structural solver *SIMPACK* enables the consideration of aeroelastic effects caused by the fluid-structure interaction (FSI).

### 2.1 Atmospheric CFD code - E-Wind

*E-Wind* is an atmospheric CFD tool developed and used by Enercon for wind resource assessment (Alletto et al., 2018). *E-Wind* solves the steady RANS equations using the $k - \epsilon$ turbulence model. The governing equations are adapted to atmospheric conditions, e.g. complex terrain, roughness and forest (vegetation), atmospheric stability and Coriolis force (Sogachev et al., 2012) and solved using the open source code OpenFOAM (v1712) as the core solver within *E-Wind*. Since the exact boundary condition for roughness and stability are often unknown, the roughness scaling factor ($RSF$) and heat flux ($HF$) are used to calibrate the simulations against mast measurements to fit the shear at the met mast location. For a detailed description see Adib et al. (2021).

### 2.2 Unsteady CFD solver - FLOWer

The basis for the numerical simulations of the wind turbine is the CFD solver *FLOWer*, which was originally developed by the German Aerospace Center (DLR) (Kroll et al., 2000). It is a compressible, block structured RANS solver. The numerical scheme is based on a finite-volume formulation. The implemented Chimera technique allows the use of independent grids for the individual components of the wind turbine and the background. The solver has been continuously extended at the author's institute to improve its suitability for wind turbine simulations. Among others, the fifth-order weighted essentially non-oscillatory scheme WENO is available for spatial discretisation (Kowarsch et al., 2013) and several hybrid RANS/LES schemes have been implemented in *FLOWer* (Weihing et al., 2018). Furthermore, a body forces approach is included to superimpose atmospheric turbulence on the inflow (Schulz et al., 2016) and forest regions are accounted for by volume forces added to the momentum equation of the Navier-Stokes equations (Letzgus et al., 2018). The work of Klein et al. (2018) introduced a revised coupling to the multi-body simulation tool *SIMPACK*.

### 2.3 Structural solver - SIMPACK

*SIMPACK* is a commercial software for the simulation of multi-body systems. The dynamic systems can consist of rigid and flexible bodies connected by joint elements. The flexible turbine components such as the tower and blades can be modelled either as beams or as modal bodies by reading in the modal properties. External forces such as aerodynamic forces can be defined internally or imported from other programs via a predefined interface environment. Controllers can also be integrated. *SIMPACK* has recently been used by industrial and research groups for the simulation of wind turbines, e.g. Luhmann et al. (2017) and Guma et al. (2021).



## 3 Computational setup

The complex terrain in Perdigão with its double ridge was simulated without and with a turbine on site. In addition, two
reference simulations were conducted in *FLOWer* with the same turbine in flat terrain.

The setup of the complex terrain simulation aimed to reproduce a measured flow situation. The situation was selected based
on operating data from the turbine in Perdigão, with the objective of having fairly constant operating conditions close to rated
conditions. This was found to be the case for a thirty-minute interval on 10 May 2017 from 15:15:00 UTC with an inflow from
southwest ($230°$). The measured data was averaged over this interval.

### 3.1 Atmospheric precursor simulation with E-Wind

*E-Wind* provides a site- and situation-specific mean flow field, as it is calibrated with mast measurements on the real site in
Perdigão. The equations in *E-Wind* were discretised using a mixed 1st/2nd order scheme (Alletto et al., 2018). Due to the
high resolution and up to date terrain and forest maps, no roughness calibration was performed ($RSF = 1$). For the selected
situation, good calibration results could be obtained for met mast 20 under neutral thermal conditions ($HF = 0$).

#### 3.1.1 Mesh and boundary conditions for E-Wind

*E-Wind* uses a cylindrical domain with a diameter of $22.5\,\mathrm{km}$ and a height of $6\,\mathrm{km}$ (see Fig. 1). The Perdigão terrain mesh in
*E-Wind* was based on the high-resolution ($2\,\mathrm{m}$) map provided by Palma et al. (2020). It was resampled to a resolution of $10\,\mathrm{m}$
to fit the mesh resolution and to avoid artefacts in the mesh. Since the scanned area is smaller than the computational domain,
the map was extended with data from the Shuttle Radar Topography Mission (SRTM). The coordinate system used for both the
maps and the domain is ETRS89 / Portugal TM06 (EPSG:3763). To allow for homogeneous inflow, the terrain was flattened
towards the lateral boundaries by a $3\,\mathrm{km}$ wide ramp followed by a $2\,\mathrm{km}$ wide flat area.

The structured mesh for *E-Wind* with about $9.4$ million cells was created with *Pointwise*. A fine grid with $10\,\mathrm{m}$ horizontal
resolution and $1\,\mathrm{m}$ vertical resolution is used in the centre of the domain, which is coarsened towards the lateral and upper
boundaries. At a height of $500\,\mathrm{m}$ above the highest terrain elevation, the horizontal resolution is doubled to $20\,\mathrm{m}$. The mesh
is aligned with the wind direction at hub height of $WD = 230°$. The ground surface is modelled as no-slip wall, using wall
functions for the turbulent quantities and a fixed heat flux for the temperature. For the upper boundary, slip conditions for
velocity, temperature, $k$ and $\epsilon$ are applied, while the pressure gradient is fixed to prescribe the geostrophic forcing. At the sides
of the domain, an *inletOutlet* boundary condition is used with a precomputed profile for the inflow and a zero gradient for
the outflow. For more details, see Alletto et al. (2018). The geostrophic wind speed was set to $11\,\mathrm{m\,s^{-1}}$, the geostrophic wind
direction to $242°$.

#### 3.1.2 Vegetation in E-Wind

To account for vegetation in *E-Wind*, a map of the roughness length $z_0$ must be provided, which is then applied as either
roughness or as forest depending on the actual $z_0$ value. The provided forest height map with $2\,\mathrm{m}$ resolution (Palma et al.,



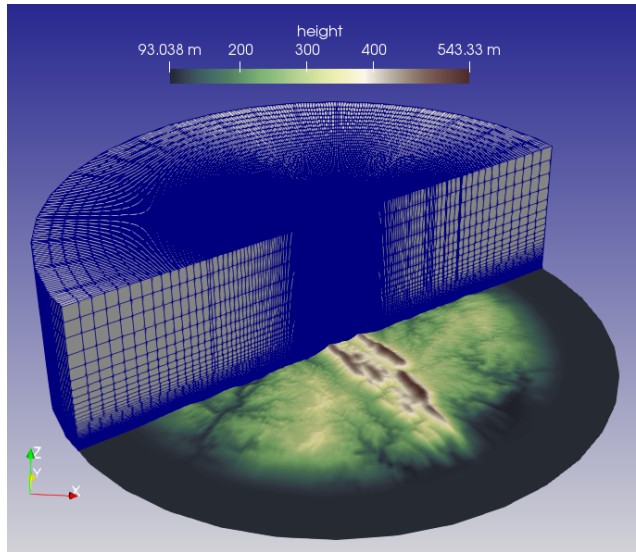

**Figure 1.** Mesh of *E-Wind* for a wind direction of $230°$ (inflow parallel to cutting plane).

2020) was resampled to $10\,\mathrm{m}$ and smoothed with a Gaussian filter. The forest height was divided by the forest scaling factor
($FSF = 10$) to derive the $z_0$ map. The calculated $z_0$ values were then classified into eight different classes. The forest model
described in Alletto et al. (2018) was applied for $z_0 > 0.5\,\mathrm{m}$, which applies to the three highest $z_0$ classes. The constant leaf
area density $LAD = 0.2$, the drag coefficient $c_d = 0.15$ and the forest height $h = FSF \cdot z_0$ were used. The lower five classes
are treated as roughness with a wall function (*nutkAtmRoughWallFunction*). A constant value of $z_0 = 1.0\,\mathrm{m}$ is used outside of
the provided map, representing forest with $h = 10\,\mathrm{m}$.

## 3.2   Complex terrain simulation with FLOWer

The unsteady simulations with *FLOWer* were carried out as DDES (Spalart et al., 2006) based on the Menter SST $k-\omega$ RANS
model (Gritskevich et al., 2013). The flow was considered to be fully turbulent. An implicit second-order dual time-stepping
scheme was deployed for time integration. The second-order Jameson-Schmidt-Turkel (JST) scheme was used for the spatial
discretisation in the boundary layer (BL) cells and the fifth-order WENO scheme was applied to the Perdigão terrain mesh to
reduce the dissipation of vortices. A physical time step corresponding to $\Delta t \approx 0.5\Delta_0/u_{hub}$ was used, where $\Delta_0$ is the smallest
grid size and $u_{hub}$ is the flow velocity at the turbine position at hub height from *E-Wind*. The dual-time stepping scheme used
80 inner iterations, which decreased the global root-mean-square density residual by two orders of magnitude. To dissipate
pressure and density disturbances caused by the enforced velocity profiles of the Dirichlet boundary condition at the inlet and
to obtain an initialisation of the mean velocity field, a pre-run with an increased time step ($\approx 70 \cdot \Delta t$) was utilized.



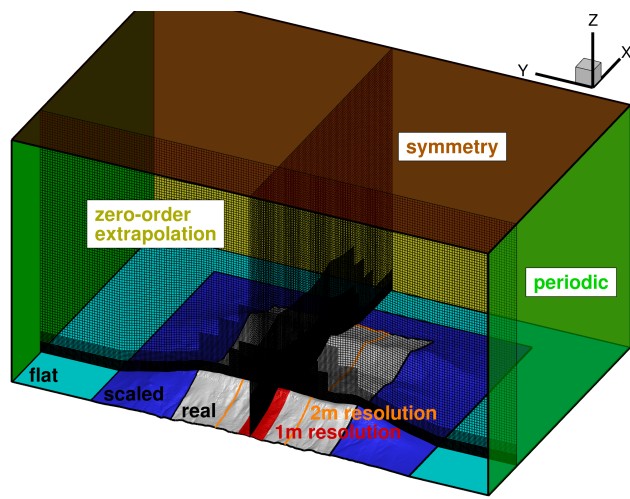

**Figure 2.** CFD domain for the *FLOWer* simulation of the complex terrain in Perdigão with boundary conditions and terrain mesh for a wind direction of $230°$.

### 3.2.1 Terrain mesh and boundary conditions for FLOWer

Basis of the Perdigão terrain mesh for *FLOWer* is the same highly resolved ($2\,\text{m}$ resolution) digital terrain model (DTM) of the site in Perdigão (Palma et al., 2020) as for *E-Wind* (without resampling). This DTM was shifted so that the tower base of the turbine is located at $x = y = 0$ and the $x$-axis was rotated to align with the horizontal wind direction $WD = 230°$ at the turbine position at hub height. To reduce the impact of the domain boundary on the flow field in the region of interest, they were placed far away ($-768\,\text{m} < x < 3072\,\text{m}$, $-3072\,\text{m} < y < 3072\,\text{m}$). In addition, the terrain was manipulated in the direction of the lateral boundaries and outlet to have a flat bottom, as visualised in Fig. 2. This allows periodic boundary conditions (BC) to be used laterally and, due to the associated reduced flow gradients, problems with numerical stability can be avoided. The domain inlet was placed at the base of the first ridge and was carried out as Dirichlet BC. The domain extends vertically up to $z = 3447\,\text{m}$. The simulated area above the ground is thus about ten times the maximum height difference of the terrain, which allows for a symmetry BC at the top (Koblitz, 2013). A zero-order extrapolation was applied at the outlet.

The Perdigão terrain mesh was created using cubic cells with a resolution of $\Delta_0 = 1\,\text{m}$ around the turbine and its direct inflow ($-768\,\text{m} < x < 512\,\text{m}$, $-160\,\text{m} < y < 160\,\text{m}$, $z < 256\,\text{m a.g.l.}$). The cells are slightly stretched and squeezed in $z$-direction and skewed to follow the terrain. This resolution is sufficient to resolve the ambient turbulence with an integral length scale $L > 20\Delta_0 = 20\,\text{m}$, following Kim et al. (2016). This region of interest is embedded in a band ($y = \pm448\,\text{m}$) with $2\,\text{m}$ resolution that covers both ridges (see Fig. 2). A coarsening of the mesh towards the domain boundaries was applied using hanging grid nodes to reduce the number of cells and to dissipate the turbulence towards the boundary conditions for stability reasons. Close to the ground (no-slip wall BC), BL cells with reduced resolution in $z$-direction (growth rate of 1.12) were included to ensure $y^+ < 5$ in the region of interest. This is crucial since without BL cells the Menter SST $k - \omega$ RANS model remains in $k - \epsilon$ mode even in the first wall normal cells (switch to $k - \omega$ only for $y^+ < 70$ (Leschziner, 2015)). Moreover, the





DDES shielding fails without BL cells and the modelled stresses are depleted, which can lead to grid induced separation on the ridge. Overall, the Perdigão terrain mesh consists of 242 million cells.

### 3.2.2 Vegetation in FLOWer

Menke et al. (2019b) discussed how sensitive the simulation result is with respect to the forest parametrisation. They found that the standard forest often used in simulations ($h = 30\,\text{m}$) causes too much drag. For this reason, much effort was put into

190 an accurate representation of the forest. In *FLOWer* the model of Shaw and Schumann (1992) is applied, which is based on the expression

$$F_i(z) = -c_d\,LAD(z)\,|u_i|\,u_i\,.\tag{1}$$

The drag force $F_i$ depends on the two forest characteristics $c_d$, a constant drag coefficient, and the leaf area density profile $LAD(z)$. Moreover, it scales with the local flow velocity $u_i$ squared.

The drag coefficient $c_d$ was set to $0.15$ as proposed by Shaw and Schumann (1992). The $LAD$ profile which characterizes the tree type was calculated by means of the leaf area index $LAI$ following Lalic and Mihailovic (2004).

The leaf-area distribution can be defined as

$$LAD(z) = LAD_m \left(\frac{h - z_m}{h - z}\right)^n exp\left[n\left(\frac{h - z_m}{h - z}\right)\right]\tag{2}$$

with

$$200 \quad n = \begin{cases} 6 & 0 \leq z < z_m \\ 0.5 & z_m \leq z \leq h \end{cases}\tag{3}$$

where $h$ is the tree height and $LAD_m$ the maximum value of $LAD$ at the corresponding height $z_m$.

The tree height $h$ and the leaf-area index $LAI$ are included in the available maps for the site in Perdigão (Palma et al., 2020). The tree types growing in Perdigão are eucalyptus and pines (Mann et al., 2017). Lalic and Mihailovic (2004) show that their model fits measured leaf-area density distributions of pine forests well with $z_m/h = 0.6$. This value was therefore applied in

this study. The maximal leaf-area density value $LAD_m$ could finally be calculated by inserting Eq. (2) into

$$LAI = \int\limits_0^h LAD(z)\,dz\,.\tag{4}$$

The contour plot of the tree height in the area upstream of the turbine for $WD = 230°$ in Fig. 3a shows many small clusters with different heights. The implementation of the forest model in *FLOWer* (Letzgus et al., 2018) expects separate forest meshes when the forest characteristics change in flow direction making it unsuitable for small clusters. Therefore, the small clusters

were summarized into six forest patches with different heights and $LAD$ distributions which were included in the simulations. The chosen patches and their tree height, which is the average value over the patch, are depicted in Fig. 3b. The $LAI$ was also averaged over each forest patch and the resulting $LAD$ distributions in Table 1 were used, respectively.



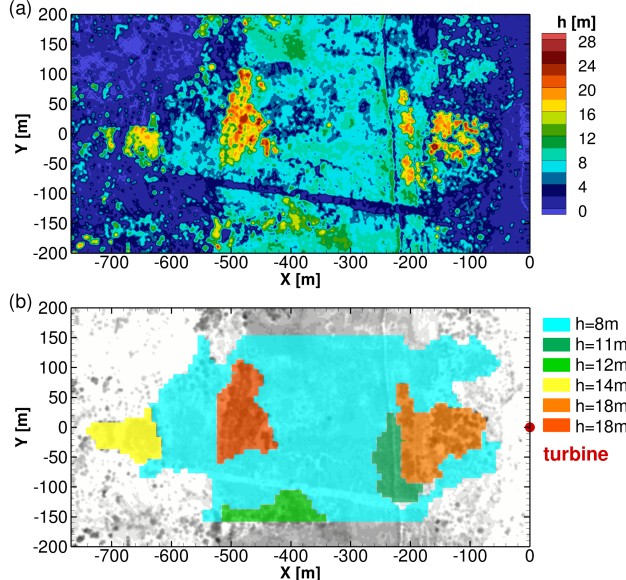

**Figure 3.** Tree height in the area upstream of the turbine for a wind direction of $230°$ (a) and forest patches included in simulation (b).

**Table 1.** Leaf-area index $LAI$, maximal leaf-area density $LAD_m$ and corresponding height $z_m$ for each forest patch.

| Forest patch | $LAI$ | $LAD_m$ [$\mathrm{m^2m^{-3}}$] | $z_m$ [m] |
|---|---|---|---|
| 8 m (cyan) | 2.59 | 0.56 | 4.8 |
| 11 m (dark green) | 2.93 | 0.46 | 6.6 |
| 12 m (light green) | 2.10 | 0.30 | 7.2 |
| 14 m (yellow) | 2.44 | 0.30 | 8.4 |
| 18 m (light orange) | 3.31 | 0.32 | 10.8 |
| 18 m (dark orange) | 3.18 | 0.30 | 10.8 |

Colours refer to Fig. 3b.

### 3.3 Atmospheric-Aerodynamics interface

The steady-state velocity profiles from *E-Wind* were passed to *FLOWer* via Dirichlet BC and the resolved synthetic turbulence
based on statistical turbulence data from *E-Wind* was superimposed.

From the results of the *E-Wind* simulation, a slice was extracted at the position of the *FLOWer* domain inlet (perpendicular to
$WD = 230°$). The three velocity components (longitudinal $u$, lateral $v$ and vertical $w$), the turbulence kinetic energy $k$ as well
as the rate of dissipation $\varepsilon$ were averaged in lateral direction ($\pm 100\,\mathrm{m}$ relative to the turbine) for each height above ground.
These values were used to create the inflow for *FLOWer*.



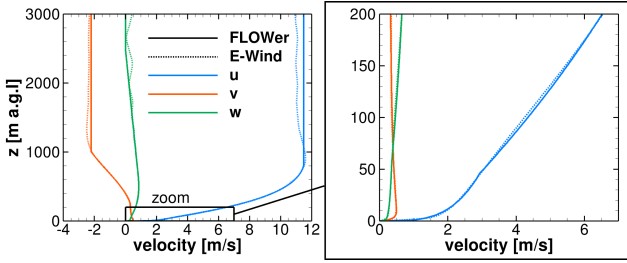

**Figure 4.** Mean velocity profiles at the *FLOWer* inlet for the entire height of the domain and zoomed to near ground level. Extracted from *E-Wind* as well as the approximation for *FLOWer* input.

Figure 4 shows the the velocity profiles of all three components above ground level (a.g.l.). The profiles of all velocity components were approximated by piecewise-defined functions such that they match the results of *E-Wind* and are constant ($u$, $v$) or zero ($w$) towards the upper boundary. Wind shear ($u(z)$), wind veer ($v(z)$) and flow inclination ($w(z)$) were prescribed at the domain boundary as Dirichlet boundary condition with the found equations. The inlet of the simulation with *FLOWer* is already in the uneven terrain at the base of the first ridge (inclination $\approx 6°$). Therefore, it is crucial to prescribe the vertical velocity component in order not to overestimate the speed-up at the ridge by deflecting the flow too much.

The turbulence intensity $TI$ of the inflow for the generation of synthetic turbulence was assessed from the laterally averaged *E-Wind* $k$ values in $77\,\mathrm{m}$ height above ground, which corresponds to the hub height, according to

$$TI = \frac{\sqrt{\frac{2}{3}k_{z=77\,\mathrm{m}}}}{\sqrt{u_{z=77\,\mathrm{m}}^2 + v_{z=77\,\mathrm{m}}^2}}\,, \tag{5}$$

with the corresponding turbulent length scale $L$

$$L = 0.09^{4/3}\frac{k_{z=77\,\mathrm{m}}^{3/2}}{\varepsilon_{z=77\,\mathrm{m}}}\,. \tag{6}$$

The resolved atmospheric turbulence for the *FLOWer* simulation was created using Mann's model (Mann, 1994) and was injected using a momentum source term (Troldborg et al., 2014), superimposing the steady sheared inflow at a distance of $L$ from the inlet. To comply with the atmospheric conditions according to IEC61400-1 (International Electrotechnical Commission et al., 2005), an anisotropic turbulence was generated. As recommended by Mann (1998), the stretching factor in the model was chosen as $\Gamma = 3.9$ to approximate the Kaimal spectral model. It was ensured that the dimension of the Mann box is larger than $8L$ in all directions and that its resolution is smaller than $L/8$.

The injection via forces as well as the numerical dissipation due to the resolution of the meshes cause a certain turbulence decay within the CFD simulation. This effect was taken into account by applying a scaling factor $f_{CFD} = 1.4$ on the velocity fluctuations of the Mann box, following Kim et al. (2016) for a propagation distance of approximately $20L$.





**Table 2.** Definition of *FLOWer* simulation cases with their inflow conditions and computational size.

| Case name | Terrain | Turbine | $u(z)$ | $v(z)$ | $w(z)$ | $u_{ref}$ [m s$^{-1}$] | $TI$ [%] | $L$ [m] | $N_{cells}$ [$10^6$] |
|---|---|---|---|---|---|---|---|---|---|
| *empty* | Perdigão | - | profile | profile | profile | 3.71 | 26.5 | 28.25 | 242.1 |
| *terrain* | Perdigão | I82 | profile | profile | profile | 3.71 | 26.5 | 28.25 | 301.2 |
| *turbulent* | flat | I82 | profile | 0 | 0 | 10.16 | 10.2 | 30.51 | 93.3 |
| *uniform* | flat | I82 | $u_{ref}$ | 0 | 0 | 10.09 | - | - | 67.3 |

### 3.4 Turbine

The examined wind turbine is a generic $2\,$MW turbine named I82 (Arnold et al., 2020) with aero-servo-elastic similarity to the commercial turbine at the site. The turbine has a hub height of $77\,$m and a rotor radius of $R = 41\,$m. The blades are pre-bended ($-1.8\,$m at the tip) and feature winglets. The rotor is mounted with a tilt angle of $5°$ and a pre-cone angle of $0°$. The tower has a bottom diameter of $d = 4.3\,$m and a top diameter of $d = 2.0\,$m.

#### 3.4.1 CFD model in FLOWer

The unsteady *FLOWer* simulations of the turbine were based on the process chain established by Klein et al. (2018).

The CFD model of the I82 turbine for the simulation with *FLOWer* consists of 17 independent meshes, which were embedded in the Perdigão terrain mesh or a flat background mesh, respectively. Three blade tip refinements and a rotor disk refinement comprise the turbine component meshes, namely lower tower, upper tower, nacelle, hub, blade-hub connectors ($3\times$), blades ($3\times$) and winglets ($3\times$). There are no gaps between the turbine components (see Fig. 5) and the boundary layer of all components is fully resolved ($y^+ < 1$). The blades were meshed in an O-topology based on the guidelines of Vassberg et al. (2008), with a special focus on a good resolution of the boundary layer and the blade wake. Three differently resolved blade meshes were used to conduct a grid convergence study following Roache (1994) ($y^+ < 1$ was kept in all blade meshes). The conservative numerical error for the medium blade mesh ($GCI^{21}_{coarse}$) is $0.4\,\%$ for thrust and $0.6\,\%$ for torque. This is acceptable, and hence the blade mesh with medium resolution was chosen, with 192 cells in radial direction, 304 cells around the airfoil, 64 cells on the trailing edge and 144 cells wall normal resulting in 11.4 million cells per blade. The growth rate in the boundary layer is 1.09. The second-order JST scheme was used for spatial discretization in the component meshes. The numerical settings of the complex terrain simulation were kept, but the time step was reduced to correspond to $1°$ azimuth for evaluation.

The background meshes for the reference simulations with flat terrain in *FLOWer* were approximately 50 rotor radii ($R$) long ($12.5R$ upstream of the rotor plane) and were approximately $25R$ wide and high, once with no-slip wall and BL cells and once with slip wall and no BL cells at the bottom. The overall number of cells $N_{cells}$ per setup is given in Table 2.





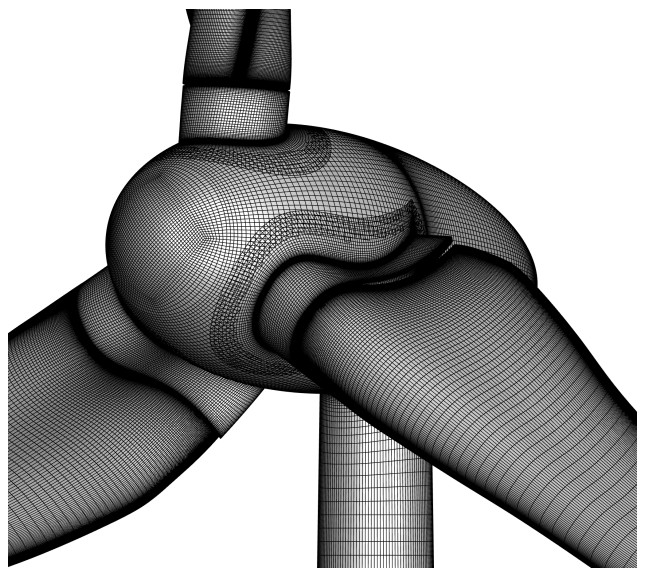

**Figure 5.** CFD surface mesh of hub region, showing mesh overlappings.

### 3.4.2 Structural model in SIMPACK

The structural model of the I82 turbine in *SIMPACK* was adopted from Arnold et al. (2020). The blades were modelled as non-
linear *SIMBEAM* using 29 flexible beam elements (Euler-Bernoulli) per blade with Rayleigh damping. The tower was adjusted
to a steel tower and modelled as linear *SIMBEAM* by using 77 flexible beam elements (Euler-Bernoulli) with a modal damping
of $\zeta = 0.002$. All eigenfrequencies below $15\,\mathrm{Hz}$ were considered in *SIMPACK*. Hub, nacelle, drive train and foundation were
defined as rigid bodies. The centrifugal force induced by the blade rotation and the gravity force were considered. The time
integrator *SODASRT_2*, a variable step-size integrator, was used to ensure that at each time step all model states were kept
within predefined tolerances.

### 3.4.3 Fluid-structure coupling

An explicit coupling scheme is applied between *SIMPACK* and *FLOWer* with both solvers running in a sequential way. After
each physical time step, information is exchanged, with *SIMPACK* using the aerodynamic loads of the previous time step to
calculate the deformations. The communication is realized by means of files containing deformations or loads at a total of 106
marker positions, of which 29 markers are allocated to each blade, 17 markers to the tower, and one marker each to the nacelle
and hub. The surface mesh is reduced to a point cloud that deforms according to the markers. The cells of the volume mesh
are linked to the point cloud via radial basis functions and thus also deform accordingly. Further details can be found in Klein
et al. (2018) and a validation of the *FLOWer-SIMPACK* coupling with an elastic cantilever beam in Klein (2019).





## 3.5  Simulation cases

To validate the wind field simulated with *FLOWer*, the complex terrain in Perdigão was simulated without turbine, as described
in Sect. 3.2. This simulation is referred to as the *empty* case. The evaluation started after the simulation of $300\,\mathrm{s}$, which were
necessary to propagate the imposed turbulence through the domain.

This *empty* case was also the basis of the simulation with the turbine I82 in the complex terrain, referred to as *terrain* case.
The turbine with its multiple component meshes was integrated into the Perdigão terrain mesh with the simulated turbulent
terrain flow (after $300\,\mathrm{s}$). 16 revolutions without fluid-structure coupling were simulated with a time step corresponding to $2°$
azimuth to reduce the disturbances due to the integration and to develop the turbine wake. Then two revolutions were simulated
with FSI to obtain the quasi steady deformation of blades and tower. A calibrated artificial damping was applied to attenuate the
starting oscillations due to the uninitialised structural model. For the evaluation simulation, the artificial damping was switched
off and the time step was reduced so that it corresponds to $1°$ azimuth.

The turbine was simulated with a constant rotational speed of $n = 16.87\,\mathrm{rpm}$, which corresponds to the thirty-minute average
(10 May 2017 from 15:15:00 UTC) of the turbine's rotational speed at the site. The corresponding pitch angle of the generic
I82 is $\beta = 4.06°$.

The two reference simulations with *FLOWer* of the I82 in flat terrain used the same operating conditions and were also run
as coupled simulations. One simulation, referred to as *turbulent* case, had an inflow with atmospheric turbulence and shear.
It was created using the method described in section 3.3. However, for this case, the *E-Wind* results were taken from a slice
at the turbine position at the top of the ridge. Hence, the effects of orography and vegetation on the horizontal wind speed
$WS$, turbulence intensity $TI$ and length scale $L$ were only included in the *FLOWer* input to the extent that *E-Wind* was able to
reproduce them. The occurring vertical velocity component at the turbine position was neglected in this setup.

The second reference simulation, referred to as *uniform* case, had an uniform inflow. The wind velocity was taken from the
*E-Wind* result on hub height and was $10.09\,\mathrm{m\,s^{-1}}$.

All evaluated simulations with turbine, including the reference simulations, were run with fluid-structure coupling. A com-
parison with a rigid turbine was not considered, as the findings from Klein et al. (2018) already show that the blade-tower
interaction, a key mechanism investigated in the following, is dominated by the blade-tower distance, which is massively
reduced when the aeroelasticity of the blades is taken into account.

The four simulation cases with the respective inflow conditions are summarized in Table 2, with $u_{ref}$, $TI$ and $L$ as reference
values at the inlet on $77\,\mathrm{m}$ a.g.l..

## 4  Results

The aim of the simulation chain is to model surface pressure fluctuations under realistic operating conditions in complex terrain
in Perdigão. Therefore, the simulated terrain flow without turbine (*empty* case) was validated first, followed by the evaluation
of the simulated turbine (cases *uniform*, *turbulent*, *terrain*).



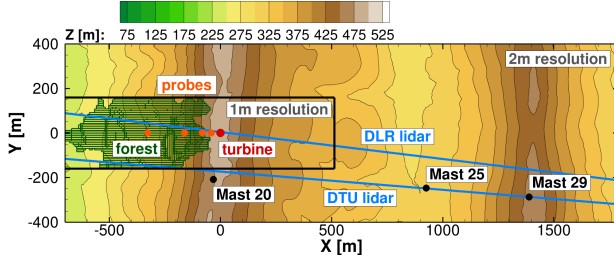

**Figure 6.** Location of lidar planes, met masts and probe positions in Perdigão relative to the turbine and properties of the *FLOWer* setup.

**Table 3.** Comparison of *E-Wind* result and *FLOWer* result ($180\,\mathrm{s}$ averaged) with met mast data ($30\,\mathrm{min}$ averaged).

| Met | $z$ | \multicolumn{4}{c}{*E-Wind*} | \multicolumn{4}{c}{*FLOWer*} |
| | | $\Delta WS$ | $\Delta w$ | $\Delta WD$ | $\Delta TI$ | $\Delta WS$ | $\Delta w$ | $\Delta WD$ | $\Delta TI$ |
| mast | [m a.g.l.] | [ms$^{-1}$] | [ms$^{-1}$] | [°] | [%] | [ms$^{-1}$] | [ms$^{-1}$] | [°] | [%] |
|---|---|---|---|---|---|---|---|---|---|
| 20 | 20 | 0.2 | −0.4 | 1.4 | 1.1 | 0.3 | −0.4 | 5.0 | −9.3 |
| | 60 | 0.1 | −0.4 | 1.1 | 0.7 | −0.2 | −0.1 | 6.8 | −7.7 |
| | 100 | 0.0 | −0.8 | 0.4 | 2.2 | 0.2 | −0.7 | 3.7 | −4.7 |
| 25 | 20 | −2.0 | 0.2 | 50.1 | 51.2 | −1.7 | 0.3 | 77.2 | 35.2 |
| | 60 | −1.6 | 0.3 | 55.8 | 27.7 | −2.2 | 0.6 | 57.1 | 17.7 |
| | 100 | −0.2 | −0.1 | 40.7 | −3.1 | −1.6 | 0.2 | 31.5 | −4.5 |
| 29 | 20 | −0.7 | −0.1 | 5.0 | −6.9 | 0.0 | 0.1 | −1.6 | −15.0 |
| | 60 | −0.6 | 0.2 | 8.2 | −6.7 | −1.1 | −0.4 | 8.2 | −11.5 |
| | 100 | −0.2 | 0.3 | 2.3 | −9.2 | −1.8 | 0.0 | 8.6 | −5.0 |

$\Delta = Simulation - Measurement.$

## 4.1 Complex terrain flow in Perdigão

The simulation of the terrain in Perdigão was intended to reproduce the measured flow situation on 10 May 2017 from 15:15:00 UTC with an inflow from southwest ($230°$). Two lidar planes (Menke et al., 2019a; UCAR/NCAR, 2019a) and the met masts (UCAR/NCAR, 2019b) shown in Fig. 6 were used for validation ($30\,\mathrm{min}$ average).

### 4.1.1 Validation of precursor simulation

Figure 7 shows the difference of the horizontal wind velocity $u_h$ in the DTU lidar plane between the *E-Wind* result and the measurement. The $x$-axis describes the distance $D$ from mast 20 in the lidar plane. Only minor differences are observed in front of the first ridge. At mast 20, which was used for calibration, a very good agreement between simulation and measurements for $WS$, $WD$ and $TI$ could be achieved (see Table 3).



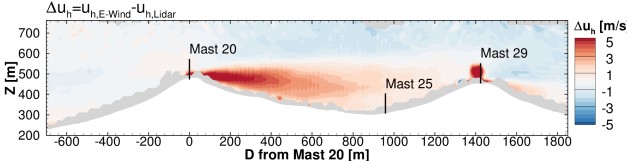

**Figure 7.** Comparison of the horizontal wind velocity $u_h$ in the DTU lidar plane between the *E-Wind* result and the measurement.

The recirculation zone behind the first ridge is underpredicted in *E-Wind*. The main reason is probably the smoothing of the terrain due to the mesh resolution of $10\,\text{m}$. Satellite images of the site show rocky terrain at the top of the ridge that could trigger early flow separation. Since these terrain features are not resolved in *E-Wind*, the flow stays attached to the ground longer and separates later, resulting in a smaller recirculation region. In the valley, large differences between simulation and measurements could be observed, especially for the lower heights. For mast 25, $WS$ is underestimated, $WD$ is off by about
$50°$ and $TI$ is overestimated in the simulations. It is well known that RANS models have difficulties in accurately predicting the size of and the flow within a recirculation zone. Furthermore, there may be some microscale or other physical effects that are not modelled in *E-Wind* (e.g. anabatic winds). For mast 29 at the second ridge, a good agreement with the measurements was observed.

### 4.1.2    Validation of simulated unsteady flow field

The unsteady results of the detailed DDES with *FLOWer* of the wind field in Perdigão without turbine (case *empty*) were averaged for $180\,\text{s}$ (equivalent to three times the duration of the Mann turbulence box) and compared with measured data. Figure 8 shows the simulated mean horizontal wind velocity in the DTU lidar plane $u_h$ of the *empty* case and the difference to the measurement. The speed-up at the first ridge agrees very well with the measurement, which is important to simulate the inflow to the turbine correctly. The velocity in the recirculation zone was captured much better than in the steady-state
precursor simulation with *E-Wind*, but still the differences in the valley between the ridges are the largest.

A comparison with the met mast data in Table 3 confirms this. At mast 20, both the horizontal wind speed $WS$ and the wind direction $WD$ fit very well over the entire mast height, indicating that the shear was also met. The vertical velocity $w$ and thus the flow inclination is slightly underpredicted. The simulated turbulence intensity $TI$ is too low because mast 20 is not located in the finest mesh region (see Fig. 6). Therefore, the flow at this position has undergone stronger numerical dissipation than
the direct inflow to the turbine. In fact, in the region with $1\,\text{m}$ resolution $2R$ in front of the turbine $77\,\text{m}$ a.g.l., the simulated turbulence intensity is about $4\,\%$ higher than at the same height at mast 20 in the simulation. Taking this offset into account, the simulated $TI$ agrees much better with the measured value on the first ridge. The slightly larger deviation at lower heights could be due to the lack of forest wake in the simulation at mast 20. Mast 25 in the valley shows large differences especially in the wind direction. This is due to a thermally driven valley flow (Fernando et al., 2019), whose physical source was included
neither in the *FLOWer* simulation nor in *E-Wind*. This missing flow also causes the wind speed to be too low and thus the $TI$ to be too high. On the second ridge at mast 29, the agreement between simulation and measurement is better again. The too low





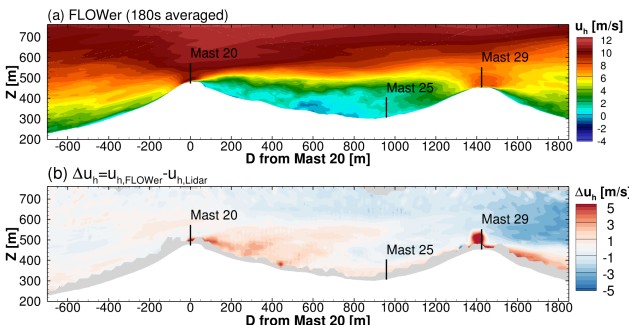

**Figure 8.** Horizontal wind velocity $u_h$ in DTU lidar plane, simulated with *FLOWer* (a) and comparison to measurement (b).

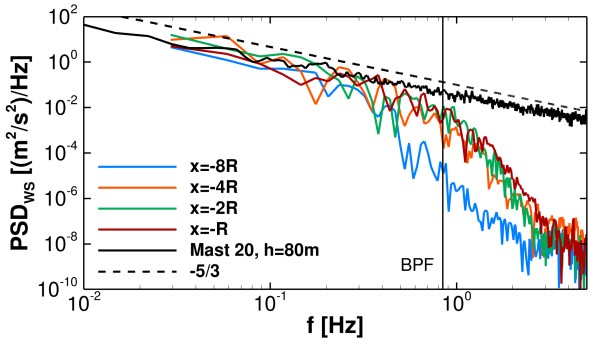

**Figure 9.** Development of power spectral density of horizontal wind speed upwind of the turbine in $77\,\mathrm{m}$ a.g.l. in *FLOWer*.

mean velocity, which can also be seen in Fig. 8, could be due to a still too small distance of the second ridge to the outlet BC in the simulation, where not fully dissipated vortices can cause a backward inflow. The lack of vegetation behind the first hill together with the increasing numerical dissipation with propagation distance in the simulation resulted, as expected, in much

too low simulated $TI$ values.

The energetic vortices in the simulation of the flow over the terrain in Perdigão were identified by calculating the power spectral density (PSD) using Welch's method with Hann window (amplitude corrected), $66\,\%$ overlap and three segments. The horizontal velocities at four positions $77\,\mathrm{m}$ a.g.l. in the direct inflow of the turbine position (see probes in Fig. 6) were evaluated. Figure 9 shows the spectra compared to the measurement at mast 20 at a height of $80\,\mathrm{m}$ a.g.l., where the energy

cascade is proportional to $f^{-5/3}$ as given by Kolmogorov for the inertial subrange. The simulation resolves this energy cascade for more than an order of magnitude before numerical dissipation causes a drop for frequencies $f > 1\,\mathrm{Hz}$. The accurately resolved part of the spectrum covers the relevant load range for wind turbines, since the blade passing frequency (BPF) for the turbine integrated into the terrain later in the simulation corresponds to $0.84\,\mathrm{Hz}$. Further up the ridge, turbulence with higher frequencies is resolved as the flow accelerates, which agrees with Spalart (2001) who found that the smallest eddies resolvable

with DES occur with a frequency of $f_{max} \approx u \cdot (5 \cdot \Delta_0)^{-1}$.





It can be concluded that the DDES of the complex terrain in Perdigão with *FLOWer* using the inflow generated from *E-Wind* results provides an accurate site- and situation-specific mean flow field as well as resolved turbulence up to $f \approx 1\,\text{Hz}$ for the evaluated situation.

### 4.2 Turbine in complex terrain

The results of the *FLOWer* simulation with the fluid-structure coupled I82 turbine at Perdigão (case *terrain*) were evaluated for 16 revolutions after initialising the wake and the deformations as described in Sect. 3.5. The flow field around the turbine, its global loads and deflections as well surface pressure fluctuations on blades and tower were investigated. The results were compared with the reference simulations in flat terrain (cases *uniform* and *turbulent*) when appropriate.

#### 4.2.1 Impact on global terrain flow

The DLR lidar was in-plane with the turbine and orientated almost parallel to the mean flow of this situation (see Fig. 6) and is therefore well suited to evaluate the impact of the turbine on the terrain flow. Figure 10a shows the difference between the *FLOWer* simulation with resolved turbine (case *terrain*) and the simulation without turbine (case *empty*). The mean horizontal wind speed $u_h$ in the lidar plane was averaged over 16 revolutions and the corresponding $60\,\text{s}$ from the *empty* simulation, respectively. The upwind induction zone in front of the turbine is relatively weak, while the wake is quite distinct up to 375 $\approx 340\,\text{m}$ behind the turbine. The wake does not follow the terrain but drifts slightly upwards. This fits well with the findings of Wildmann et al. (2018) from measurements under neutral stratification, however the velocity deficit decays faster in their study. A comparison with the measured mean horizontal velocity for the selected situation (see Fig. 10b) also shows a faster decay. It should be noted, however, that the lidar plane has an offset of $\approx 7°$ from the mean wind direction, which leads to a drift of the wake out of the measured plane. In the simulation, the flow is unintentionally more aligned with the DLR lidar 380 plane, which is evident from the offset $\Delta WD$ found for mast 20 in Table 3. Moreover, the too low $TI$ in the simulation causes the wake to be slightly too stable. Overall, the influence of the turbine on the terrain flow is well captured.

#### 4.2.2 Global Loads and deflections

Figure 11 shows the blade tip displacement out of the rotor plane $\Delta x_{oop}$ of one blade for all cases. The *uniform* case shows a clear sinusoidal trend mainly caused by the gravity load and the rotor tilt. The blade deformation overcompensates the pre-385 bend, so gravity contributes more to the out-of-plane displacement when the blade is pointing upwards. The impact of the blade-tower interaction, and hence the unsteady aerodynamic loads, on the blade deformations is only weakly recognisable by the faster decrease of $\Delta x_{oop}$ shortly after the tower passage.

The inclusion of turbulence in the simulation massively increases the amplitude of fluctuations that occur once per revolution. This is due to the large variations in wind speed over the rotor disk arising from shear and turbulence. In both cases, 390 turbulence predominates over shear effects, which can be seen in the irregular pattern. The turbulent eddies are smaller than the rotor disk ($L < 2R$) and therefore cause load oscillations with the rotation frequency. Moreover, the rotational periodicity is



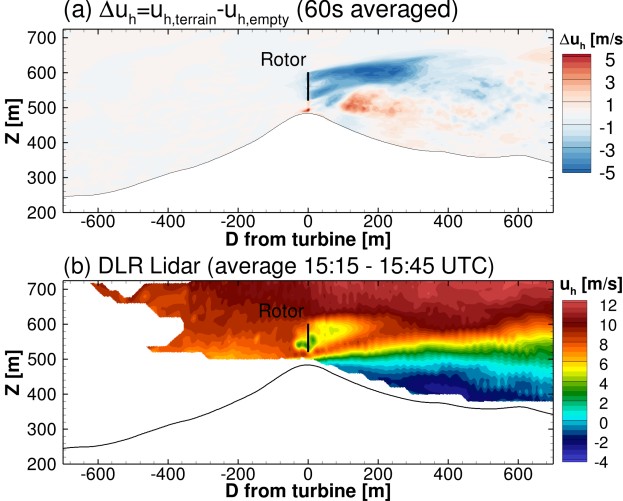

**Figure 10.** Difference of horizontal wind speed $u_h$ in the DLR lidar plane between the *FLOWer* results of the simulation with and without turbine (a) and the measured $u_h$ (b).

superimposed by stochastic broadband fluctuations caused by very small eddies. The out-of-plane deformations mostly follow the aerodynamic blade bending moment $M_y$. However, a comparison between the *turbulent* and the *terrain* case shows that the inflow turbulence cannot be generalised and has a very unique impact on the behaviour of the turbine in each case. This illustrates how important it is to model the inflow realistically and site specific. The small difference in the mean blade tip deformation $\overline{\Delta x}_{oop}$, given in Table 4, is due to differences in the global loads caused by slightly different flow conditions at the turbine position. The local inflow to the turbine is characterised by a mean velocity $U_{ref}$ and a mean flow inclination angle $\gamma$ one $R$ in front of the turbine at hub height. Table 4 summarizes the flow at the turbine position for the three cases and lists the different extracted mean powers $\overline{P}$ and mean acting thrusts $\overline{F}_x$.

The inflow generation described in Sect. 3.5 was intended to result in similar loads at the turbine. However, it turned out that the underlying *E-Wind* results overestimated the velocities at the turbine position. Due to a recirculation zone that was too small, the streamlines in *E-Wind* followed the terrain too closely and were thus more curved, which led to too high an acceleration. The unsteady aerodynamic effects analysed in the following are not significantly altered by differences in mean loads and can still be compared between the cases.

### 4.2.3 Surface pressure on tower

The surface pressure $p$ and its distribution are the dominant source of the aerodynamic loads and are evaluated on the tower in the following. In many respects, be it fatigue loads or acoustic emissions, the fluctuations and the distribution of the acting forces are of greater importance than the magnitude of the steady load. Figure 12 shows contour plots of the standard deviation $\sigma$ of the surface pressure on the tower for all cases. In the plots the angle $\Phi$ on the $x$-axis corresponds to the circumferential position of the tower, where $180°$ is the upwind side where the blades pass. Three main areas of interest can be distinguished,





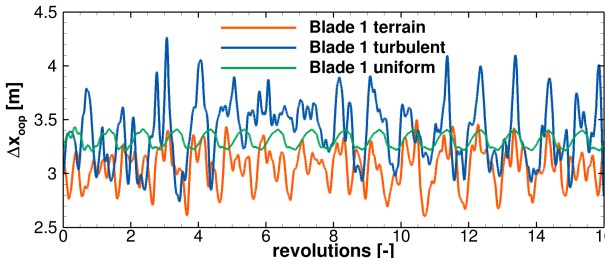

**Figure 11.** Deformation of blade tip out of the rotor plane $\Delta x_{oop}$ for all cases.

**Table 4.** Local inflow to the turbine and global loads on the turbine for all simulation cases with turbine.

| Case name | $U_{ref}$ | $\gamma$ | $\overline{\Delta x_{oop}}$ | $\overline{P}$ | $\overline{F_x}$ |
|---|---|---|---|---|---|
| | [m s$^{-1}$] | [°] | [m] | [MW] | [kN] |
| *uniform* | 9.2 | 0.2 | 3.31 | 1.5 | 221 |
| *turbulent* | 9.5 | −1.6 | 3.33 | 1.6 | 230 |
| *terrain* | 8.9 | 13.6 | 3.06 | 1.2 | 196 |

marked with horizontal dotted lines in red (①, ②, ③). Below the blade tip passage (①), inflow turbulence increases the fluctuations, especially in the *terrain* case, while at the height of the blade tip passage (②), the fluctuations are actually reduced in both cases with turbulent inflow compared to uniform inflow. The main effect of the blade on the tower is at around $50\,\text{m}$ height (③) on the side of the descending blade ($\Phi \approx 210°$) for all cases. The cause of these observations is examined
separately for each height in the following.

Below the blade passage on height ①, the time series of the surface pressure fluctuations $p-p_{avg}$, where $p_{avg}$ is the local time average, on a line around the tower were extracted and plotted as contour plots in Fig. 13. The uniform inflow causes distinct, periodic patterns, especially on the back and crosswind sides of the tower ($120° > \Phi > 240°$), which increase in intensity over time. The two cases with turbulent inflow, on the other hand, are dominated by larger patterns that are less regular. Nevertheless,
the *turbulent* case develops a fine pattern on the tower back after some time. For the *terrain* case, patterns are by far the largest. Opposing pressure fluctuations occur at the tower sides, which remain stable for multiple revolutions and then swap.

With a transformation to the frequency domain using Welsh's method (compare Sect. 4.1.2), the observed pattern can be better characterised. The PSDs in Fig. 14 show that for uniform inflow, the main fluctuations occur at the tower sides and back at discrete frequencies that are not multiples of the BPF. These pressure fluctuations can be associated with a periodic
separation known as the Kármán vortex street. However, according to Horvath et al. (1986), the local Reynolds number of the tower $Re = 2.5 \cdot 10^6$ falls into the supercritical regime, where vortex shedding can occur over a wide range of frequencies and is quite unstable or even not observed at all in some experiments. Nevertheless, they found two dominant shedding frequencies in their experiments for supercritical $Re$-numbers. This fits well with the observation in Fig. 14a with two dominant frequencies





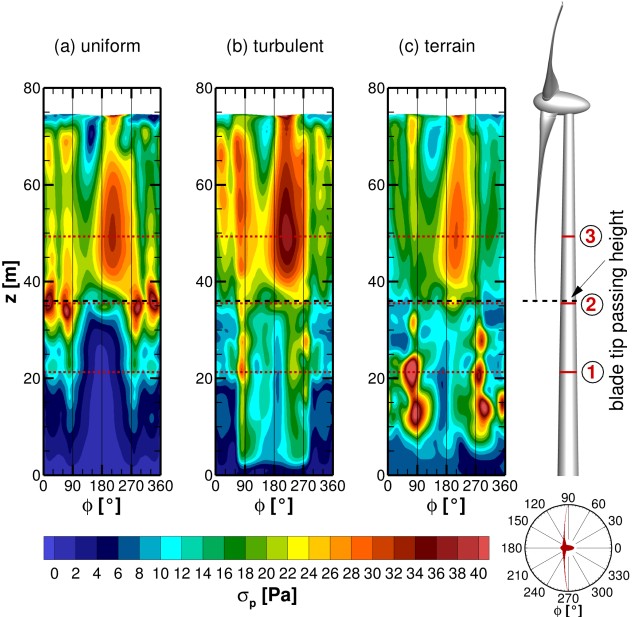

**Figure 12.** Standard deviation $\sigma$ of surface pressure $p$ on tower for all cases (a)-(c).

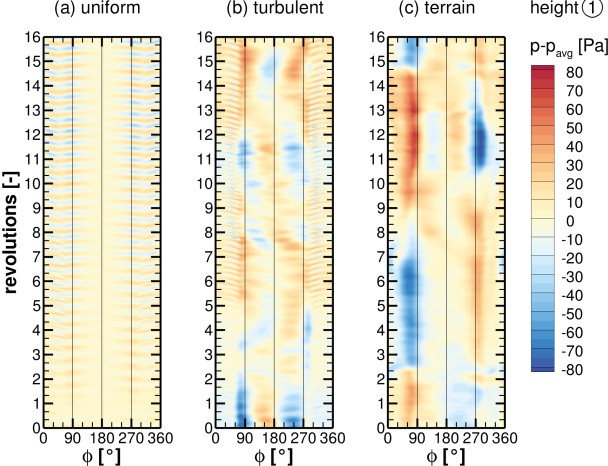

**Figure 13.** Time series of surface pressure fluctuations $p - p_{avg}$ on a line around the tower on height ① for all cases (a)-(c).

at $f = 0.59\,\mathrm{Hz}$ and $f = 1.55\,\mathrm{Hz}$. Considering the time history in Fig.13, it can even be stated that the shedding characteristic
changes over time, which underlines the instability of the vortex street.

Figure 15 shows the instantaneous vortex structures visualised with the $\lambda_2$-criterion, coloured with the vertical component of vorticity $\omega_z$. With uniform inflow, coherent vortex cells with constant shedding frequency form and extend over the entire tower height. This phenomena is well known for tapered cylinders (e.g. Johansson et al. (2015)), although it is remarkable that





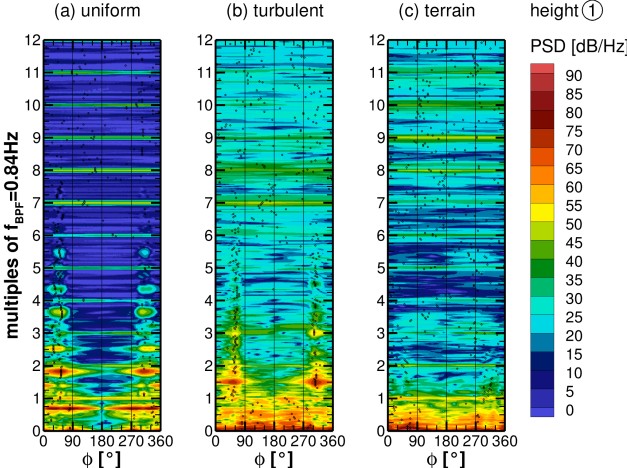

**Figure 14.** Power spectral density of surface pressure fluctuations on a line around the tower on height ① for all cases (a)-(c).

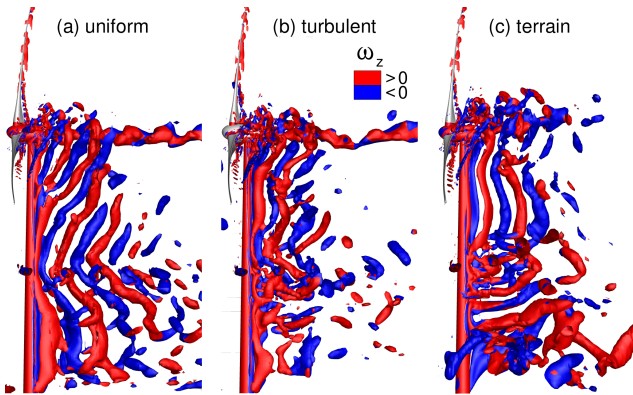

**Figure 15.** Vortex structures after 16 revolutions visualised with $\lambda_2$-criterion and coloured with vertical vorticity $\omega_z$ for all cases (a)-(c).

only one vortex cell forms over the entire tower height, not even broken up by the blade tip vortices (not shown). Therefore,

using the local tower diameter to calculate the Strouhal number of the shedding frequencies is not appropriate. Using the mean tower diameter gives $St = fd/u = 0.18$, which fits the experimental results of Jones Jr. (1968), and $St = 0.48$, which is similar to the higher eddy-shedding frequency measured by Horvath et al. (1986) and simulated by Rodríguez et al. (2015).

Turbulence in the inflow hampers the periodic vortex shedding on height ①, as shown by the reduction of discrete frequencies in the PSD for the *turbulent* case and an absence of discrete frequencies in the *terrain* case in Fig. 14. The vortex structures

in Fig. 15b and 15c in the lower tower section confirm this. Especially in the *terrain* case, rather horizontal, streamwise vortex structures tend to occur at the tower and the coherence in the vortex shedding is suppressed in lower tower regions. This vortex shape is triggered by the terrain flow in two different ways. First, the sheared acceleration of the mean flow $\Delta u$ due to the slope of the ridge rotates and stretches the vertical vorticity $\omega_z$ into streamwise vortices with increased $\omega_x$, as can be seen in Fig. 16.





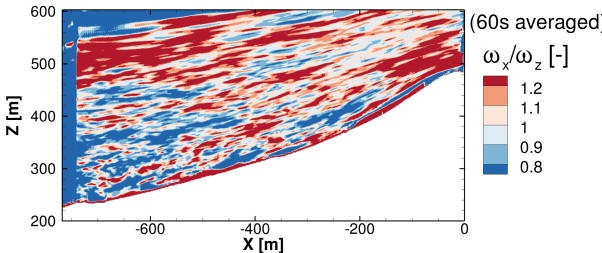

**Figure 16.** Ratio of horizontal and vertical vorticity $\omega_x/\omega_z$ in a slice ($y = 0$) upstream of the turbine.

Belcher and Hunt (1998) found that $\omega_x \sim \Delta u$ for turbulent flow over the top of a hill. Second, the ridge near the separation
point in front of the turbine is not smooth in the crosswind direction, but has bumps similar in shape to the wedges used for
passive flow control in aviation or automotive. Such obstacles give rise to strong streamwise vortex structures (McCullough
et al., 1951) that interact with the flow around the lower tower in the *terrain* case. The streamwise vortices are very stable and
the side changes observed in Fig. 13c are presumably triggered by corresponding temporary changes in the wind direction in
the direct inflow.

More general, Batham (1973) also found that turbulence suppresses coherent vortex shedding and Bruun and Davies (1975)
reported a reduction in vortex shedding correlation length for turbulent flow, both for critical Reynolds numbers. This shows
that the consideration of realistic inflow conditions alters the occurring physical phenomena considerably. Generically simpli-
fied setups carry the risk of enhancing stable patterns, which can lead to overestimated tonalities in acoustic evaluations, for
example.

For both cases with turbulent inflow, the energetic inflow turbulence (compare Fig. 9 for *terrain* case) dominates at frequen-
cies far below the BPF at height ①, as visible in Fig. 14b and c. For all cases, the BPF and its higher harmonics are faintly
visible in the PSDs even at this height.

The evaluation of the pressure fluctuations at height ②, where the blade tips pass, results in the pressure curves over time in
Fig. 17 and the PSDs in Fig. 18. With uniform inflow, the pattern of pressure fluctuations in Fig. 17 is very constant over time.
The fluctuations at the back of the tower are much stronger than at height ①, while at the tower front ($\Phi \approx 180°$) additional,
very sharp periodic fluctuations occur. The inflow turbulence in the *turbulent* and *terrain* case clearly changes the pattern also
at this height. Compared to the lower height ①, a finer periodic pattern is noticeable, which occurs especially at the tower
front.

Almost all around the tower, but particularly at the tower front, pressure fluctuations with the BPF and its harmonics are
clearly visible for all cases at height ② in Fig. 18. They are imposed by the blade tip vortices periodically hitting the tower
with the BPF and sweeping over its circumference. Since this periodic interaction is very brief it acts as an impulse on the
tower and many higher harmonics are visible in the PSD.

Looking at Fig. 18a for uniform inflow, it can be seen that the strongest fluctuations still occur with a frequency $f = 0.59\,\text{Hz}$
at the tower sides and back. The amplitude of these pressure fluctuations is even higher than at height ①, since the vertically



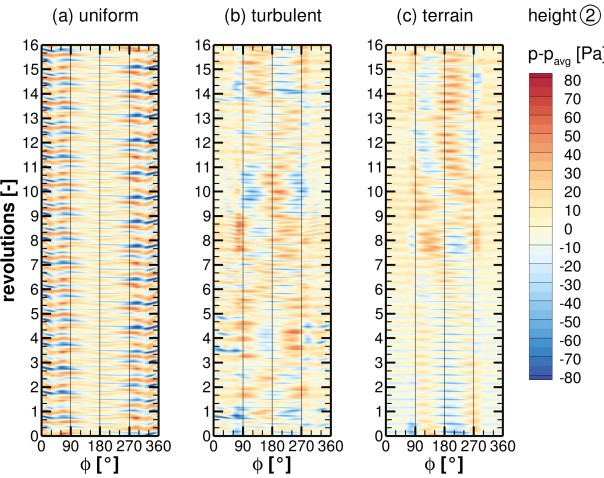

**Figure 17.** Time series of surface pressure fluctuations $p - p_{avg}$ on a line around the tower on height ② for all cases (a)-(c).

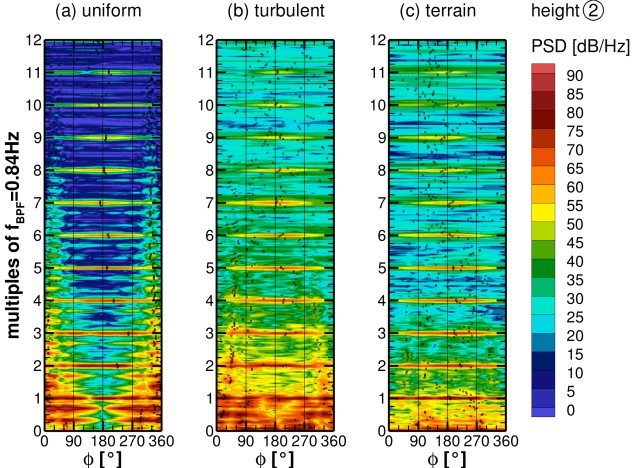

**Figure 18.** Power spectral density of surface pressure fluctuations on a line around the tower on height ② for all cases (a)-(c).

stretched shed vortices have their highest vorticity in the middle part, where the local tower diameter corresponds to the mean tower diameter.

For the *turbulent* case, strong pressure fluctuations still occur at the tower sides/back below BPF associated with vortex shedding, but no discrete shedding frequency can be identified in Fig. 18b. Instead, the inflow turbulence imposes strong broadband pressure fluctuations around the whole tower for frequencies below BPF. The PSD of the *terrain* case in Fig. 18c

looks remarkably different below BPF. This is because the terrain flow causes quite different vortex structures at height ②, which is evident when comparing Fig. 15b and 15c. As described, the inflow turbulence in the terrain flow is much more anisotropic, with $\omega_z$ being converted to $\omega_x$, and streamwise vortices cause less pressure fluctuations on the tower surface.



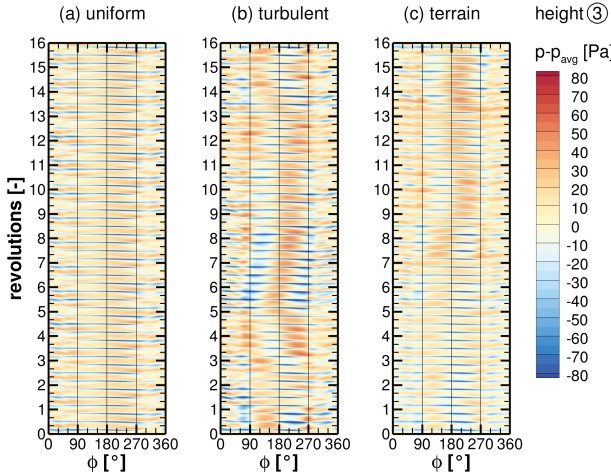

**Figure 19.** Time series of surface pressure fluctuations $p - p_{avg}$ on a line around the tower on height ③ for all cases (a)-(c).

Moreover, the turbulence intensity in the near-field of the turbine is not identical between the *turbulent* and the *terrain* case in this study, with $TI$ being $1.5\%$ lower in the *terrain* case. These two factors explain the lower broadband fluctuations in the
*terrain* case.

At height ③ the blade passage causes very sharp periodic pattern on the side of the descending blade ($\Phi \approx 210°$) for uniform inflow, as visible in Fig. 19a. The turbulent cases also show this periodic pattern (see Fig. 19b and 19c), but less sharp and superimposed by low frequency patterns. A less strong periodic pattern on the back of the tower is also visible for all cases, indicating vortex shedding with discrete frequencies again.

Pressure fluctuations with discrete frequencies of the BPF and its harmonics have the highest amplitudes in all cases at height ③, shown in Fig. 20. The fluctuations with the BPF dominate around the whole tower since the reduced pressure on the suction side of the blade extends around the whole tower when the blade passes. The strongest fluctuation with BPF occur on the side of the descending blade ($\Phi \approx 210°$), marked with black symbols in Fig. 20. This was also found by Klein (2019) and is due to a speed up of the flow between tower and the approaching blade, known as Venturi effect, locally enhancing the
pressure reduction. For the higher harmonic pressure fluctuations of the BPF the maxima slightly drift towards the tower front as the frequencies increase.

With uniform inflow, even at height ③ where the blades pass, the same vortex shedding frequency is pronounced at the tower sides and back as at the lower heights, as visible in the PSD in Fig. 20a. This confirms that coherent vortex cells stretch over the entire tower height for uniform inflow, even with the blade wake interaction and a tapered shape of the cylinder. The
curved shape of the vortex cells in Fig. 15a is due to the reduced flow velocity behind the blades caused by the blade induction, resulting in a slower downwind propagation of the vortices.

The *turbulent* case shows the same vortex shedding frequency, but much less pronounced, with a more broadband character of the pressure fluctuations below BPF. At height ③, the *terrain* case shows vortex shedding for the first time with a fairly



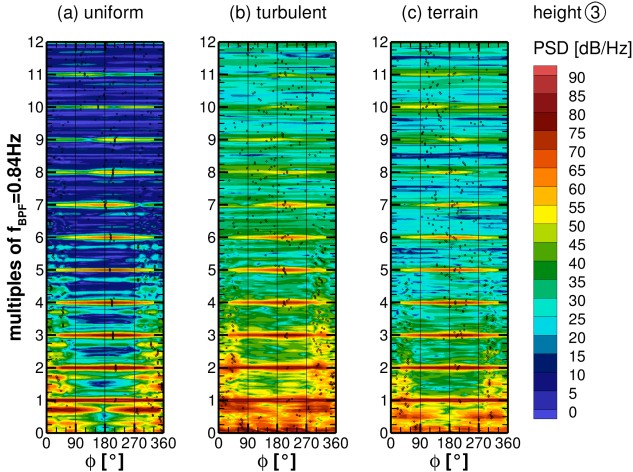

**Figure 20.** Power spectral density of surface pressure fluctuations on a line around the tower on height ③ for all cases (a)-(c).

discrete frequency at the tower back, however, at $f = \mathrm{BPF}/2$. Figure 15c shows the coherent vortices at the upper tower. As

mentioned, the horizontal inflow vortices prevent the formation of strong vortex cells extending over the entire tower height and thus the blade passage impulse is dominant enough to induce a periodic vortex shedding on the upper tower, one vortex per blade passage with opposite circulation. This interaction between blade passage and vortex shedding was also described by Gómez et al. (2009), who performed two-dimensional simulations of the blade-tower interaction.

Figure 21 shows the maximum amplitude of the pressure fluctuations on the tower with the BPF ($0.84\,\mathrm{Hz}$) and its first

two higher harmonics ($1.69\,\mathrm{Hz}$ and $2.53\,\mathrm{Hz}$) and the circumferential position $\Phi$ where they occur. Behind the blade passage, above $z = 35\,\mathrm{m}$, neither the position nor the amplitude of the strongest pressure fluctuations with BPF or the first two higher harmonics are significantly altered by the different inflow conditions. This means that the mechanisms of the blade-tower interaction remain unchanged.

The observations made lead to the conclusion that the surface pressure fluctuations on the tower are dominated by a super-

position of blade-passing effects and tower vortex shedding, as also described by Klein et al. (2018). The inflow turbulence itself has no significant influence on the fluctuations at the tower, however, it is shown how crucial it is to take it into account realistically nevertheless, since it massively alters the vortex shedding characteristic and thus the periodicity of the surface pressure fluctuations which can drive the emergence of acoustic low frequency tonalities.

### 4.2.4   Surface pressure on blades

Besides the tower, the blades are the turbine components with the largest surface area, which makes them relevant low-frequency emitters. Moreover, they are the components with highest aerodynamic loads. Figure 22 shows contour plots of the standard deviation $\sigma$ of the surface pressure on one blade for all cases. In the plots the blade is unwound and the arc length $d$ from the leading edge (LE) is normalized with the local chord length $c$, where positive values belong to the suction side (SS)



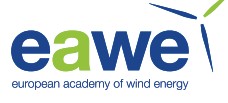


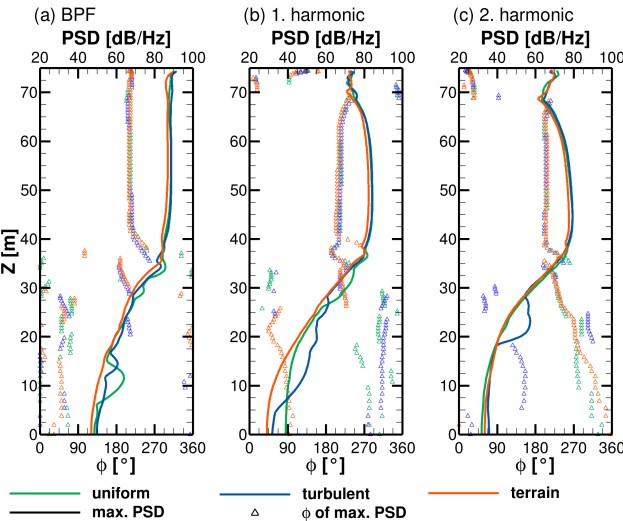

**Figure 21.** Maximum amplitude of the PSD of the surface pressure fluctuations per height and their circumferential position on the tower for BPF (a), first (b) and second (c) higher harmonic.

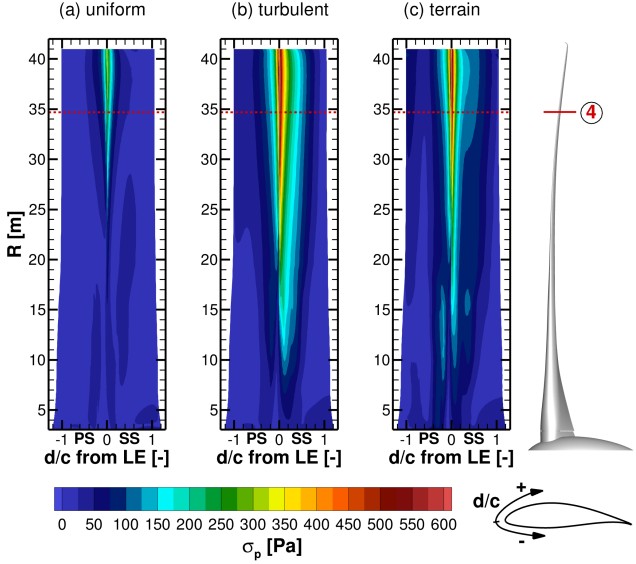

**Figure 22.** Standard deviation $\sigma$ of surface pressure $p$ on one blade for all cases (a)-(c).

and negative values to the pressure side (PS) respectively. Pressure fluctuations are the strongest close to the LE for outer blade
radii. Inflow turbulence significantly increases the fluctuations and broadens the region in both *turbulent* and the *terrain* case.
To further look into details a position ④ at $85\,\%$ blade radius marked with the red line was chosen. At this radial position the
blade generates the highest thrust per meter.



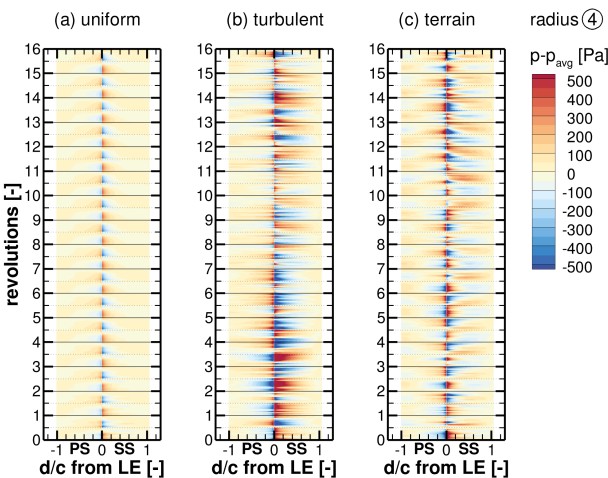

**Figure 23.** Time series of surface pressure fluctuations $p - p_{avg}$ along a blade section at radius ④ for all cases (a)-(c).

The time series of the pressure fluctuations $p - p_{avg}$ at the blade radius ④ in Fig. 23 show a periodic pattern over the whole circumference with a frequency of once per revolution for the *uniform* case. Towards the LE, the fluctuations are by far the

strongest and opposite compared to the main part of the airfoil. The reversal of the pressure pattern between descending (from full to half revolution) and ascending (from half to full revolution) blade is due to the rotor tilt. It causes the effective angle of attack $\alpha$ at radius ④ to be slightly less for the descending blade than for the ascending ($\Delta\alpha \approx 0.28°$). This leads to a small periodic shift of the stagnation point, increasing the pressure on the SS close to LE for the descending blade. In contrast, the effective inflow velocity $u_{eff}$ at the blade at radius ④ is slightly higher for the descending blade than for the ascending

($\Delta u_{eff} \approx 1.7\,\mathrm{ms}^{-1}$). This dominates the global blade load and leads to a lower pressure on the SS and higher one on the PS from $\approx 0.4c$ to the trailing edge for the descending blade. These effects also occur for the *turbulent* and the *terrain* cases, but are superimposed by the unsteady changes in local flow velocity and direction caused by the inflow turbulence, which generates additional stochastic pressure fluctuations. The coupled unsteady blade deflection (see Fig. 11) additionally changes $u_{eff}$ and $\alpha$, resulting in a complex interaction. As on the tower, the *terrain* flow causes less strong fluctuations compared to the *turbulent*

case due to the described difference in the inflow turbulence. In addition, the inclined flow reduces $\alpha$ for the ascending blade and decreases it for the descending blade, respectively. This counteracts the periodic angle of attack variation and consequently load fluctuations caused by the tilt. Furthermore, the slightly sheared inflow ($\Delta u \approx 0.5\,\mathrm{ms}^{-1}$ over the rotor) in these two cases marginally increases the blade loads in the upper half of the revolution, reducing the pressure on SS and increasing it on PS. For all cases, the tower passage causes a very sharp, impulsive disruption of the pressure pattern by a sudden reduction in $\alpha$

due to the reduced flow velocity in front of the tower and due to the acceleration of the flow between blade and tower, known as the Venturi effect. In addition, the higher pressure in the tower dam region is imposed on the blade.

A transformation into the frequency domain, depicted in Fig. 24, confirms the observations. For the *uniform* case at the blade radius ④, peaks are visible at the rotational frequency $f_t = 0.28\,\mathrm{Hz}$ and its multiples. The peak at $f_t$ is caused by a


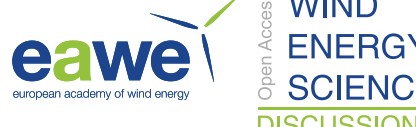

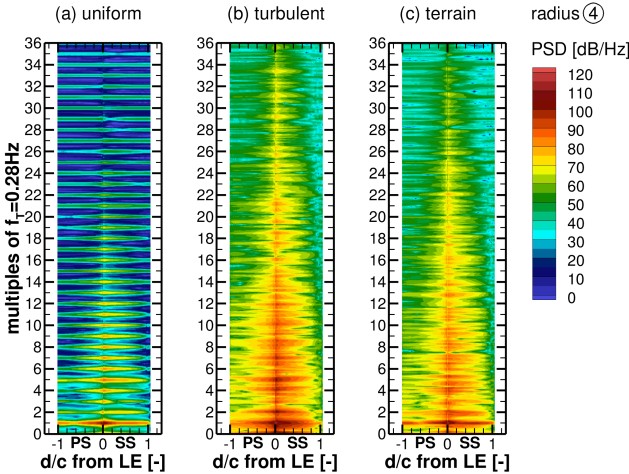

**Figure 24.** Power spectral density of surface pressure fluctuations on a line around the blade at radius ④ for all cases (a)-(c).

combination of the tilt effect and the blade-tower interaction. The tilt effect is sinusoidal and therefore the higher harmonics
are caused solely by the impulsive blade-tower interaction. The PSD also shows that the pressure fluctuations are not limited
to the LE but occur around the whole blade, which is difficult to see in Fig. 23a. Inflow turbulence in the *turbulent* and the
*terrain* case dominates at the blade radius ④ above $f_t$ and masks the higher harmonics caused by the blade-tower interaction,
resulting in a broadband characteristic of the pressure fluctuations.

The most pronounced pressure fluctuations occur in all inflow cases at the rotational frequency. However, the maximum
amplitude for that frequency is slightly stronger in the *turbulent* case than for uniform inflow due to the shear effect, which is
also sinusoidal, and is further amplified in the *terrain* case by the inclination effect.

The observations made show that the surface pressure fluctuations on the blade are dominated by a superposition of design
properties, such as blade-tower distance and rotor tilt, which determine the blade-tower interaction, and inflow properties, such
as turbulence characteristic and flow inclination. The inflow turbulence causes broadband fluctuations, the intensity of which
is directly related to the turbulence characteristics. Therefore, it is important to take the inflow into account realistically. The
impact of the blade-tower interaction on the blade is only slightly altered by the inflow, though.

## 5  Conclusions

In this paper, results of a detailed turbine simulation coupled with an atmospheric code were presented to analyse the character-
istics of blade-tower interactions as a cause of low-frequency noise emissions under complex terrain inflow. A highly resolved
computational setup for a DDES of the complex terrain in Perdigão, including vegetation, has been established. Guidelines
are given for dimensions and computational settings to obtain numerically stable results, and limitations in simulating valley
flow are shown. A new workflow for the generating site- and situation-specific inflow conditions using a steady atmospheric
precursor simulation with *E-Wind* was presented. By calibrating this simulation with met mast data, a real situation on 10 May



2017 was simulated and the flow field was investigated. A validation with met mast and lidar data showed that a site- and situation-specific flow field in Perdigão can be simulated well and with high accuracy using *FLOWer* as main solver, especially with respect to the local flow conditions at the turbine position.

In a second step, a fully resolved generic wind turbine coupled to a structural solver was included in the simulation of the complex terrain in Perdigão. Due to its aero-servo-elastic similarity to the real turbine at the site, the expansion of the turbine wake could also be compared with lidar measurements and was found to be simulated similarly.

The accurate simulation of the flow field around the turbine in Perdigão allows a realistic evaluation of the unsteady impact of the flow in complex terrain on surface pressure fluctuations on the turbine. Two reference simulations in flat terrain, one with uniform inflow and one with generic inflow turbulence, were performed to highlight differences. It was shown that turbulence as it occurs in the complex terrain in Perdigão massively alters the frequencies and position of strong surface pressure fluctuations caused by vortex shedding on the tower, also compared to turbulent inflow in flat terrain. This is due to the streamwise stretching of the turbulence in the inflow caused by the acceleration at the ridge and terrain characteristics. However, the dominance of the pressure fluctuations with the BPF and its higher harmonics at the upper tower section, caused by the blade-tower interaction, is not noticeably altered by the more realistic inflow. At the blade, on the other hand, the pressure fluctuations with multiples of the tower passage are masked by turbulent inflow, leaving the pressure fluctuations with the rotational frequency as the only discrete frequency in an otherwise broadband regime caused by the interaction of the blade with the inflow turbulence. The specific turbulence characteristics of the terrain flow have, however, no explicit effect on the pressure fluctuations on the blade compared to the generic turbulence in flat terrain, but the flow inclination counteracts shear and tilt effects.

In future studies it is planed to post process the simulation results with a Ffowks-Williams-Hawking solver to evaluate the low frequency acoustics directly and to localise the main sources of low-frequency emissions on wind turbines.

*Author contributions.* FW created the high-fidelity *FLOWer* setup, performed the CFD-MBS simulations, did the evaluation and wrote most of the paper. JL was responsible for the atmospheric *E-Wind* simulations and contributed the related parts of the paper. TL and EK initiated the research, supervised the work and revised the manuscript.

*Competing interests.* The authors declare that they have no conflict of interest.

*Acknowledgements.* The authors are very grateful to the German Federal Ministry for Economic Affairs and Energy (BMWi) for funding the research within the framework of the joint research projects Schall_KoGe (FKZ 0324337C) and IndiAnaWind (FKZ 0325719F). The authors gratefully acknowledge the High-Performance Computing Center Stuttgart (HLRS) for providing computational resources within the project WEALoads.



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
