# Peer review of "Impact of the wind field at the complex terrain site Perdigão on the surface pressure fluctuations of a wind turbine"

_Wind Energy Science, 2021_

## Author Comment (AC2)

**Reply to comments by Reviewer Nr. 2**

Florian Wenz on behalf of the authors
IAG, University of Stuttgart

April 7, 2022

The authors would like to thank the reviewer for his/her efforts and valuable comments. They are very much appreciated and incorporated into the revised paper.

In the present document the comments given by the 2nd reviewer are addressed consecutively. The following formatting is chosen:

- The reviewer comments are marked in blue and italic.

- The reply by the authors is in black color

- A marked-up manuscript is added. Changed section with regard to the comments by reviewer 2 are marked in yellow. Changed sections with regard to comments by both reviewers are marked in grey. General changes made by the authors are marked in green.

The authors revised the English and reformulated several sentences. The tense is switched to present where possible to improve readability. In addition, the case names are changed from *uniform* to *flat+unif* and from *turbulent* to *flat+turb* to avoid confusion with the use of these words as adjectives. These changes are marked-up in green.

**General comments "G"**

1. "*The description of the scope and objectives of the paper is too limited.*"

The authors reformulated the scope and objectives to give more details on the content and state the research questions more clearly, see $\boxed{\textbf{R2:G1-a}}$ (page 4, line 106).

2. "*There is an extensive description of methods used but not why they are used. The novelty of the used approach could be further elaborated.*"

The use of DDES is motivated by the overall aim of capturing highly unsteady aerodynamic effects that can cause low-frequency noise emissions. This is made clearer in the reformulated scope and objectives (see $\boxed{\textbf{R2:G1-a}}$ (page 4, line 106)). The current state of the art suggests the use of DDES for the investigation of unsteady effects occurring at wind turbines (compare Section 1.1 (page 2)). A note on this is added in the manuscript, see $\boxed{\textbf{R2:G2-c}}$ (page 7, line 198).

The authors included a note on why a measured situation is to be reproduced (see $\boxed{\textbf{R2:G2-a}}$ (page 6, line 155)) and further explain why a precursor simulation with *E-Wind* is necessary for this (see $\boxed{\textbf{R2:G2-b}}$ (page 6, line 162)).

The consideration of FSI is essential because the blades are bend significantly during operation, which reduces the blade-tower distance and thus directly affects the observed surface pressure

on the blades and tower. A note is added to the manuscript regarding this, see $\boxed{\textbf{R2:G2-d}}$ (page 14, line 324).

To the authors' knowledge, the manuscript represents the first study with simulations of a fully resolved turbine in the complex terrain at Perdigão (compare Section 1.2 (page 3)). Furthermore, the coupled *E-Wind-FLOWer* approach proposes a new method for generating suitable inflow conditions for high fidelity CFD simulations of complex terrain.

3. "*The result section contains about 15 pages. There are many interesting results but I partly miss the connection between scope and what research questions that are addressed and conclusion. I think it would be suitable for the authors to consider if this paper could be separated into two papers. Perhaps one with the focus of the flow modelling at this site and one on the surface pressure? Or at least connect the scope, results and conclusion more clearly. Perhaps separating the discussion from the results and have a separate section on discussion could improve readability.*"

Thank you for your suggestion. The authors prefer to stick to one paper. The reason for this is that the flow field itself at this site has already been covered in several publications. This part of our paper would therefore, in our opinion, not add enough novelty to what is already known to justify a stand-alone paper. Instead, the evaluation and validation of the flow field serves to prove the quality and suitability of the process chain for the detailed simulation of a wind turbine in complex terrain. The focus of the paper should be on the interaction between rotor blade, tower and inflow, and this is where the authors see the added scientific value.

The authors have divided the results section into two separate parts for better readability, see $\boxed{\textbf{R2:G3-a}}$ (page 15, line 371) and $\boxed{\textbf{R2:G3-b}}$ (page 19, line 441). Moreover, new section headings are introduced (see $\boxed{\textbf{R2:G3-c}}$ (page 22, line 502) to $\boxed{\textbf{R2:G3-e}}$ (page 26, line 569)) and paragraphs are joined together.

Moreover, the research question is stated more clearly in the reformulated scope and objectives (see $\boxed{\textbf{R2:G1-a}}$ (page 4, line 106)) and is added more precisely throughout the manuscript to emphasis the connection, see $\boxed{\textbf{R2:G3-f}}$ (page 3, line 59) to $\boxed{\textbf{R2:G3-h}}$ (page 19, line 442).

4. "*Something went wrong in the layout of fig 4.*"

The right picture shows a zoom in of the near ground region of the velocity profiles depicted in the left plot. The plots show the velocity profiles in all three directions as extracted from *E-Wind* and as approximated for the *FLOWer* input. The authors adjusted Figure 4 (page 12) for clarification.

5. "*The figures in general could benefit from a check of font size etc.*"

All figures were checked for readability and the font size was adjusted accordingly.

[revised manuscript text omitted]

---

## Author Comment (AC3)

**Reply to comments by Reviewer Nr. 3**

**Florian Wenz on behalf of the authors**
**IAG, University of Stuttgart**

**April 7, 2022**

The authors would like to thank the reviewer for his/her efforts and valuable comments. They are very much appreciated and incorporated into the revised paper.

In the present document the comments given by the 3rd reviewer are addressed consecutively. The following formatting is chosen:

- The reviewer comments are marked in blue and italic.

- The reply by the authors is in black color

- A marked-up manuscript is added. Changed section with regard to the comments by reviewer 3 are marked in orange. Changed sections with regard to comments by both reviewers are marked in grey. General changes made by the authors are marked in green.

The authors revised the English and reformulated several sentences. The tense is switched to present where possible to improve readability. In addition, the case names are changed from *uniform* to *flat+unif* and from *turbulent* to *flat+turb* to avoid confusion with the use of these words as adjectives. These changes are marked-up in green.

**General comments "G"**

1. "*The presentation of the main conclusions from this work should be improved and made more precise, and it should be ensured that what is written in the manuscript text and abstract/conclusions is consistent. Now there are problems with this, in part because the English should be improved throughout the manuscript. The interpretation of what the authors want to convey is convoluted due to imprecise (and sometimes incorrect) formulations. For example, in the abstract*
*- "The conservation of the flow field," –> what does conservation mean here?*
*- "It is shown that a sophisticated DDES of the complex terrain plays a decisive role in the unsteady aerodynamics of the turbine, due to its specific flow characteristic" –> I am not quite sure what the authors want to precisely convey (state that DDES captures unsteady dynamics better than RANS?) here. Although it should be noted that this is not shown in the present work as the turbine dynamics are only studied in the FLOWer simulations. Also, what is "decisive," i.e., how much is the influence?*"

The authors completely reformulated the abstract (see $\boxed{\text{R3:G1-a}}$ (page 1, line 14)), the scope and objectives (see $\boxed{\text{R3:G1-b}}$ (page 4, line 106)) and the conclusion (see $\boxed{\text{R3:G1-c}}$ (page 32, line 690)) to better present the research questions and link the results to them.

In addition, the authors have clarified the conclusions drawn throughout the manuscript and formulated them more cautiously (see $\boxed{\text{R3:S12-a}}$ (page 18, line 436), $\boxed{\text{R3:G1-d}}$ (page 28, line 601) and $\boxed{\text{R3:G1-e}}$ (page 31, line 648)).

2. "*Furthermore, it is unclear the accuracy of the obtained results. Significant differences between the field observations and simulations are observed; however, after section 4.1, these are not discussed further. The case under consideration is challenging, so differences with observations by no means disqualify the importance of the study. However, please convey to the reader how they should interpret the "accuracy" of the results in section 4.2. For example, to what degree do the results depend on the numerical resolution, the domain size, and the forest modeling. Especially on the latter, the authors repeatedly say that it is essential, but there is no demonstration of the importance of forest modeling. Were the authors able to perform validations beyond the comparison of the flow fields?*"*

In the former section 4.1 (now 4.2) the authors give possible reasons for the differences in the flow field between measurement and simulation. Missing physical phenomena for the valley flow as well as the impact of the outlet boundary are mentioned. The authors have also added this to the conclusions from this section (see $\boxed{\text{R3:S12-a}}$ (page 18, line 436)). The larger differences mainly occur downstream of the turbine position, however, the authors are confident that the flow situation at the turbine position (at the top of the first ridge) is realistically captured. Therefore, these deviations are not discussed further and the load analysis in former section 4.2 (now 5) is reliable to the authors opinion.

Regarding the impact of the resolution, a grid convergence study for the component meshes is presented in Section 3.4.1 (page 13). The resolution of the terrain mesh (1m) mainly influences the size of the smallest resolved eddies and thus the highest frequency in the spectra of the inflow. In Figure 9 (page 19) the spectra is shown and eddies up to 1 Hz are well resolved. This fits the findings of Kim et al.[1] mentioned in Section 3.2.1 (page 8) who recommend 20 cells per integral length scale when a RANS/LES hybrid model is used for turbulent flows with the 5th order WENO scheme (in the present study 28 cells per length scale are used.)

A note is given on how the differences in loads are to be interpreted, see $\boxed{\text{R3:G2-a}}$ (page 21, line 487) and following sentence.

The domain size and terrain manipulation described in Section 3.2.1 (page 8) is the result of a continuous improvement of the setup by the authors with the aim of keeping the dependence of the results on the setup as low as possible.

[Figure]

Figure 1: Impact of forest in the Perdigão on wind velocity on first ridge.

Investigations on the influence of the forest on the flow field were carried out, but are not shown in the manuscript due to lack of space. Fig. 1 shows the velocity difference of simulations with and without forest (each averaged over 60s). Effects on turbulent structures can be seen, as

well as the slightly higher velocities at the turbine position in the simulation with forest due to the higher flow displacement.

A validation of the turbine loads or deformations was not possible because no load measurements were available. Furthermore, a generic wind turbine and not the commercial turbine was simulated. However, the deformations and loads were checked for plausibility to the best of the authors' knowledge.

**Specific comments "S"**

1. "*Line 31: Large-scale meteorological effects are often captured by Large Eddy Simulations (LES) with meteorological codes such as the Weather Research and Forecasting (WRF) model –> This suggests WRF is LES code, which is not the case.*"

To the authors understanding and also according to the given citation WRF also provides a LES model.

The authors reformulated the sentence, see $\boxed{\textbf{R3:S1-a}}$ (page 2, line 47).

2. "*There are various references to what seem to be "variables" or "settings" within the code (such as RSF=1; HF=0 (around line 129), FSF=10 (Line 150), nutkAtmRoughWallFunction) (lines (153), and various others so please double-check the entire text). In contrast, the user codes are not fully available. Please clarify these accordingly such that the manuscript can be fully understood by readers that do not have access to the text.*"

The variables $RSF$ and $HF$ are shortly introduced in Section 2.1 (page 5). However, the authors agree that they do not contribute to the understanding of the performed simulation and have therefore removed them from the manuscript, see $\boxed{\textbf{R3:S2-a}}$ (page 5, line 131) to $\boxed{\textbf{R3:S2-c}}$ (page 6, line 168).

The variable $FSF$ is defined in Section 3.1.2 (page 7) and is not a solver specific variable. It is the ratio between vegetation height and roughness length. It is used to derive the roughness map from the available vegetation height map.

The settings *nutkAtmRoughWallFunction* and *inletOutlet* are OpenFOAM boundary conditions which is now explicitly mentioned, see $\boxed{\textbf{R3:S2-d}}$ (page 7, line 184) and $\boxed{\textbf{R3:S2-e}}$ (page 7, line 194). As OpenFOAM is a free software, the authors consider this information useful for potential readers.

The *SIMPACK* specific settings are removed from the manuscript, see $\boxed{\textbf{R3:S2-f}}$ (page 14, line 327) to $\boxed{\textbf{R3:S2-h}}$ (page 14, line 331).

3. "*Section 3.2.2. Just above equation (2), it is stated that the LAD is calculated based on the LAI. However, it is not clear how. In fact, in equation (4), the LAI is just defined as the vertically integrated LAD. Please clarify the section correspondingly. Note that also zm is undefined.*"

The equations (2) and (3) are merged into (2) and the former (4) is now (3). Substituting (2) into (3) leaves $LAD_m$ as the only unknown variable for which the integral can be solved numerically.

The authors reformulated the description, see $\boxed{\textbf{R3:S3-a}}$ (page 9, line 241) to $\boxed{\textbf{R3:S3-c}}$ (page 10, line 252).

$z_m$ is introduced in the manuscript directly following equation (2) and is defined at $\boxed{\textbf{R3:S3-d}}$ (page 10, line 250).

4. "*Section 3.2.2. How do you select the different zones in figure 3? (note that 18m is used twice in figure 3; see Table 3).*"

The authors added a description of the selection of the patches, see $\boxed{\textbf{R3:S4-a}}$ (page 10, line 261). The process is realized with a Matlab script based on the Matlab function "alphashape". The procedure allows to obtain several separate patches with the same tree height if they are each large enough but too far apart to be merged. The summarizing Table 1 (page 10) is adjusted for clarity reasons.

5. "*In figure 2, you show that the region in front of the turbine is smoothed to ensure "periodicity of the terrain". Is the terrain smoothing not larger than adding the Forrest in that region?*"

The terrain mesh is only smoothed in the direction of the lateral boundaries from $y = \pm 1024\,\text{m}$ onwards and in the direction of the outlet from $x = +2048\,\text{m}$ onwards. In front of the turbine, the terrain is not manipulated but accurately reproduces the DTM with $2\,\text{m}$ resolution.

The authors clarified this in the manuscript (see $\boxed{\textbf{R3:S5-a}}$ (page 8, line 214)) and adjusted Figure 2 (page 8). Further links to the colored areas in Figure 2 (page 8) are added, see $\boxed{\textbf{R3:S5-b}}$ (page 9, line 221) and $\boxed{\textbf{R3:S5-c}}$ (page 9, line 225).

6. "*Section 3.2.2. Is it correct that you only model forest in part of the computational domain? Why was this choice made?*"

Yes, the forest is simulated only upstream of the turbine in the finest mesh region with $1\,\text{m}$ resolution. The authors have now mentioned this more clearly in the manuscript, see $\boxed{\textbf{R3:S6-a}}$ (page 9, line 235) and $\boxed{\textbf{R3:S6-b}}$ (page 10, line 261).

The authors decided to neglect the forest in the remaining domain because of the high preprocessing effort, especially with respect to the creation of many separate Chimera-compatible meshes (compare Fig. 2). The impact of this simplification on the flow field in the vicinity of the turbine is expected to be very small. Immediately downstream of the turbine, the terrain descends rapidly and a large recirculation zone with low velocities is formed, where the forest is likely to have little impact. In particular, feedback to the turbine position is not expected. Further downstream and laterally, the mesh is coarsened to save computational resources, and this is expected to have a much larger impact on the flow field than neglecting the forest. The focus in this manuscript is on the effects of the terrain on the aerodynamics of the turbine rather than on the accurate simulation of the global flow field throughout the complex terrain.

[Figure]

Figure 2: Tree height in the Perdigão for a wind direction of 230°.

7. "*line 222: Wind shear (u(z)), wind veer (v(z)) and flow inclination (w(z)) –> The definition of these terms is incorrect. Please ensure proper use of the terms in the entire manuscript.*"

The authors revised the use of these terms throughout the manuscript, see $\boxed{\textbf{R3:S7-a}}$ (page 5, line 132) to $\boxed{\textbf{R3:S7-j}}$ (page 31, line 642). In particular, the frequently used term "shear" has been clarified by "vertical wind shear".

8. " "laterally averaged E-Wind k values" –> Please define how k is determined in E-wind"

*E-Wind* is a steady-state RANS solver using the $k - \epsilon$ turbulence model, compare Section 2.1 (page 5). The turbulent kinetic energy $k$ and the rate of dissipation $\varepsilon$ are the two transported variables and are hence solved in the whole *E-Wind* domain.

The authors added the definition of $k$ additionally in Section 2.1 (page 5), see R3:S8-a (page 5, line 126).

The authors rephrased the corresponding paragraph in Section 3.3 (page 11), see R3:S8-b (page 11, line 285) and R3:S8-c (page 12, line 289).

9. "Equation (6) epsilon does not seem to be defined."

$\varepsilon$ represents the rate of dissipation, see R3:S9-b (page 11, line 272).

The authors added the definition of $\varepsilon$ additionally in Section 2.1 (page 5), see R3:S9-a (page 5, line 127).

10. "Figure 4: As E-Wind solution is used as inflow for FLOWer simulation. How is it possible that these "inflows" are not the same? In particular, the u-component. Can you explain how this coupling is done?"

The profiles of the laterally averaged velocity components from *E-Wind* (dashed in Figure 4 (page 12)) are approximated with piecewise-defined functions (solid in Figure 4 (page 12)), compare from R3:S10-b (page 11, line 275). Keeping $u$ and $v$ constant and $w = 0$ towards the upper BC in *FLOWer* avoids numerical instabilities. This leads to small deviations above $1000\,\mathrm{m}$ a.g.l., however, for the near-ground flow field they are almost the same (see zoom in Figure 4 (page 12)).

The authors adjusted the explanation for a better understanding, see R3:S10-a (page 11, line 270) to R3:S10-f (page 11, line 279).

11. "Table 4: How is the difference in TI defined? Percentage points or relative difference?"

The authors assume the reviewer refers to Table 3 (page 16). The difference is defined in percentage points.

The authors clarified this in the manuscript, see Table 3 (page 16) and R3:S11-d (page 17, line 413) and R3:S11-e (page 26, line 567).

12. "lined 363, see also line 565: This conclusion is quite strong, considering that a significant difference with field data is observed at various places in the domain of interest. Yes, at mast 20, the agreement is good, but at the other masts, less so. Presumable results are tuned to the mast 20 location in some way."

The authors formulated the conclusion more cautious and assessed the results in a more differentiated way, see R3:S12-a (page 18, line 436) and R3:G1-c (page 32, line 690).

As described in Section 2.1 (page 5) *E-Wind* can be calibrated with measurement data and in Section 3.1 (page 6) it is stated that Mast 20 was used in this study as calibration data. The process chain aims to have the flow at the turbine position as close to reality as possible.

13. "Improve figure quality; in particular, make sure the used font size for axis labels and legends is sufficiently large."

All figures were checked for readability and the font size was adjusted accordingly.

14. "*Figure 16: What happens at the inflow boundary at x=-700m? Why is the solution not continous there?*"

The *FLOWer* domain starts at $x = -768\,\mathrm{m}$, compare Section 3.2.1 (page 8). While the velocity profiles are prescribed at the inlet ($x = -768\,\mathrm{m}$), the turbulence is injected via forces at a distance of $L = 28\,\mathrm{m}$ from the inlet ($x = -768 + L = -740\,\mathrm{m}$), compare Section 3.3 (page 11). Therefore, the vorticity also suddenly increases at $x = -740\,\mathrm{m}$ and the plot shows a discontinuity there.

The authors added an explanation in the manuscript, see $\boxed{\textbf{R3:S14-a}}$ (page 24, line 532).

15. "*Towards the end of the manuscript, the text is not divided into proper paragraphs, but paragraphs only have 1 or 2 sentences. The manuscript's structure can be improved by collecting relevant material in relevant paragraphs/sections.*"

The authors have divided the results section into two separate parts for better readability, see $\boxed{\textbf{R3:S15-a}}$ (page 15, line 371) and $\boxed{\textbf{R3:S15-b}}$ (page 19, line 441). Moreover, new section headings are introduced (see $\boxed{\textbf{R3:S15-c}}$ (page 22, line 502) to $\boxed{\textbf{R3:S15-e}}$ (page 26, line 569)) and paragraphs are joined together.

16. "*line 569 "the expansion of the turbine wake could also be compared with lidar measurements and was found to be simulated similarly." Where is this discussed in the manuscript?*"

The authors renamed the section dealing with the turbine wake to make it easier to find, see $\boxed{\textbf{R3:S16-a}}$ (page 19, line 450).

17. "*line 577: This contrasts with what you write at 395. Please rephrase.*"

The statement in line 577 was intended to refer to the impact of inflow on the pressure fluctuations on the tower, while in line 395 it refers to the impact of the inflow on the blade loads and deformations.

The authors clarified this in the manuscript, see $\boxed{\textbf{R3:S17-a}}$ (page 20, line 477).

18. "*The manuscript would benefit from proofreading by someone with native English proficiency.*"

The authors revised the English and reformulated several sentences. The tense is switched to present where possible to improve readability. In addition, the case names are changed from *uniform* to *flat+unif* and from *turbulent* to *flat+turb* to avoid confusion with the use of these words as adjectives. These changes are marked-up in green.

**References**

[revised manuscript text omitted]
 ($-768\,\mathrm{m} < x < 3072\,\mathrm{m}$, $-3072\,\mathrm{m} < y < 3072\,\mathrm{m}$) , as visualized in Fig. 2.  **R3:S5-a** In addition, the DTM is smoothed at the lateral and rear boundaries (dark blue area) to blend into a flat bottom
215 (light blue area), while it remains unchanged in the region of interest (grey are). This allows periodic  BCs to be used laterally and, due to the associated reduced flow gradients, problems with numerical stability can be avoided. The domain inlet  is placed at the base of the first ridge and  is carried out as Dirichlet BC. The domain extends vertically up to $z = 3447\,\mathrm{m}$. The simulated area above the ground is thus about ten times the maximum height difference of the terrain, which allows for a symmetry BC at the top (Koblitz, 2013). A zero-order extrapolation  is applied at the outlet.

220  The Perdigão terrain mesh  is shown in Fig. 2. It is created using cubic cells with a resolution of $\Delta_0 = 1\,\mathrm{m}$ around the turbine and its direct inflow ($-768\,\mathrm{m} < x < 512\,\mathrm{m}$, $-160\,\mathrm{m} < y < 160\,\mathrm{m}$, $z < 256\,\mathrm{m}$ a.g.l. R3:S5-b , marked with red lines). The cells are slightly stretched  or squeezed in $z$-direction and skewed to follow the terrain. This resolution is sufficient to resolve the ambient turbulence with an integral length scale $L > 20\Delta_0 = 20\,\mathrm{m}$, following Kim et al. (2016). This region  is embedded in a band ($y = \pm 448\,\mathrm{m}$) with $2\,\mathrm{m}$ resolution that covers both ridges and resolves detailed terrain features

[revised manuscript text omitted]

---

## Author Response (AR2)

**Reply to comments by Chief Editor**

**Florian Wenz on behalf of the authors**
**IAG, University of Stuttgart**

**June 10, 2022**

The authors would like to thank the Chief Editor for his support and valuable comments. They are very much appreciated and incorporated into the revised paper.

In the present document the comments given by the Chief Editor are addressed consecutively. The following formatting is chosen:

- The Chief Editor comments are marked in blue and italic.

- The reply by the authors is in black color

- A marked-up manuscript is added. Changed section with regard to the comments by the Chief Editor are marked in yellow. Changed sections with regard to comments by reviewer 3 are marked in orange. General changes made by the authors are marked in green.

**Specific comments "S"**

1. "*Please make figure 7, 8 and 10 larger.*"

Figure 7 (page 15) and Figure 8 (page 16) are enlarged from two-column width to one-column width. Moreover the colorbar is slightly adjusted. The size of Figure 10 (page 18) is kept. However, the depicted area is reduced by zooming in on the wake. Isolines are added to improve the visibility of the wake.

2. "*In figure 9 I would really like you to plot the frequency times the power spectral density which is commonly done. The reason is that your y-scale covers an enormous number of decades which makes is difficult to see the differences between the curves clearly.*"

The PSD in Figure 9 (page 17) is multiplied with the frequency, reducing the y-axis by six decades.

3. "*I am looking forward to the application of the F-W-H procedure to get to the acoustic emmission.*"

Thank you for your interest. The first acoustic results were recently presented at the Torque Conference in Delft (DOI: 10.1088/1742-6596/2265/3/032060).

**Reply to comments by Reviewer Nr. 3**

Florian Wenz on behalf of the authors
IAG, University of Stuttgart

June 10, 2022

The authors would like to thank the reviewer for his/her efforts and valuable comments. They are very much appreciated and incorporated into the revised paper.

In the present document the comments given by the 3rd reviewer are addressed consecutively. The following formatting is chosen:

- The reviewer comments are marked in blue and italic.

- The reply by the authors is in black color

- A marked-up manuscript is added. Changed section with regard to the comments by reviewer 3 are marked in orange. Changed section with regard to the comments by the Chief Editor are marked in yellow. General changes made by the authors are marked in green.

**General comments "G"**

1. "*I believe the authors have improved the manuscript and. I previously inquired about the accuracy and validation of the simulations, and I believe that this has not been fully addressed yet. As indicated in my previous report, it is understandable when simulations and experiments do not fully agree. Based on the provided answers that seems to be the case as turbulence intensity levels are not accurately reproduced (a difference of 5 to 10 percentage points in atmospheric turbulence at the location of interest is significant). I think this should be more clearly discussed in abstract and conclusion, instead of just stating there that all agrees very well. Now the discussion of differences between experiments and simulations is somewhat limited and mainly presented in section 4.2. For comparison to the experiments and validation of the results, table 3, section 4.2, is particularly relevant. Can you please double-check/confirm that the presented differences in turbulence intensity are indeed in percentage points (with a percentage point being the unit for the arithmetic difference of two percentages. For example, moving up from 40 per cent to 44 per cent is an increase of 4 percentage points.) For the FLOWer domain, the difference in turbulence intensity is 5 to 10 percentage points at the turbine location, which is very significantly different given general atmospheric turbulence levels.*"

The authors apologize for the unclear presentation of the flow results in terms of $TI$. The difference of 5 to 10 percentage points in $TI$ at Mast 20 given in Table 3 (page 14) in the previous version of the manuscript was rather confusing as it does not give a clear indication of the actual ability of the simulation to resolve turbulence. The reason for the stated large deviation is the too coarse resolution (2m) in the FLOWer simulation at this position (see

Figure 6 (page 14)). Therefore $TI$ is completely removed from Table 3 (page 14) and the section on the validation of the turbulence is rewritten, see R3:G1-c (page 16, line 350).

The authors hope that with the new presentation of the results it will become clearer that the simulated wind field near the turbine is indeed quite close to the measurement, with $\Delta WS = 0.0\text{m/s}$, $\Delta WD = 2.8°$, $\Delta w = -0.5\text{m}$ and $\Delta TI = 2.6\text{pp.}$ at hub height. Therefore, no detailed discussion of the differences was included in the abstract and conclusion. However, the authors have added some further information in the abstract (see R3:G1-a (page 1, line 7)) and a remark on $TI$ in the conclusion (see R3:G1-d (page 30, line 595)).

For the validation of the precursor simulation with *E-Wind*, $TI$ is completely removed, since the steady-state simulation does not resolve turbulence at all, see R3:G1-b (page 14, line 321) and following.

2. "*This difference between experiments and simulations seems to relate to the transition from the Ewind solution at the entrance of the FLOWer domain from a Mann modelled turbulence spectrum to a flow that accurately captures atmospheric turbulence. In figure 16, you indicate that the flow discontinuity at the entrance of the FLOWer domain is related to this. This seems to indicate that the flow needs to transition from the Mann modelled spectrum, which can require significant space. I find the answer in the response document is clearer than what is added to the manuscript.*"

The turbulence is injected via forces 28m downstream of the inlet, so that in the first 28m there is no turbulence and consequently no vorticity in the simulation domain (as visible in Figure 16 (page 23)). The authors agree that the resolved turbulence needs a certain distance to develop from the injected forces. In the authors' experience, however, this is much less than 700m. Nevertheless, this transition is associated with considerable numerical dissipation as is the propagation over 700m. This should be compensated for with the scaling factor $f_{CFD} = 1.4$ mentioned in Section 3.3 (page 9). The too low $TI$ is due to an insufficient correction of this numerical dissipation. This is now stated in the manuscript, see R3:G1-c (page 16, line 350).

Moreover, the section on the discontinuity in Figure 16 (page 23) is rewritten, see R3:G2-a (page 22, line 466).

3. "*In the conclusion (see, for example, line 570), it is now stated that turbulence is captured well without discussing these aspects. However, a more balanced discussion addressing the aspects indicated above seems warranted. I.e. it should be clear from the abstract and conclusion that the Mann spectrum is used at the interface between FLOWer and Ewind and that this could have limitations.*"

The authors added a sentence stating the usage of the Mann model in the conclusions, see R3:G3-a (page 30, line 590). Moreover, the quality of the result in terms of $TI$ is stated, see R3:G1-d (page 30, line 595).

**Specific comments "S"**

1. "*Table 2: the naming of "empty" and "terrain" is perhaps not ideal.*"

The authors adjusted the names of these cases. The case "empty" is now called "terrain+noWT" (see R3:S1-a (page 12, line 289) to R3:S1-g (page 18, line 392)) and the case "terrain" is now named "terrain+WT" (see R3:S1-h (page 12, line 293) to R3:S1-ad (page 29, line 576)).

2. "*Table 3: It would be helpful also to present the absolute values and not just the difference. This would make it easier to assess statements like the one on 336. "At mast 20, both the*

*horizontal wind speed WS and the wind direction WD fit very well over the entire mast height, indicating that the vertical wind shear is met." –> with the information at hand, the shear (or difference in shear cannot be determined)."*

The absolute values of $WS$, $w$, and $WD$ of the measurement are now added to Table 3 (page 14). Moreover, the values for Mast 20 at 80m are added, as reference for hub height.

3. *"line 359: "It can be concluded that the DDES of the complex terrain in Perdigão with FLOWer ... ". In table 3, it is now indicated that the difference in Turbulence intensity is 5 to 10 percentage points. Given that turbulence intensity can affect the loading on turbines, this is important to discuss, see also above."*

The PSD in Figure 9 (page 17) gives a good indication of which frequencies in the inflow turbulence are captured with sufficient energy. Near the turbine ($x = -R$) this is the case almost up to the BDF and hence sufficient to get a representative loading of the turbine. A remark on $TI$ is added, see $\boxed{\textbf{R3:S3-a}}$ (page 17, line 376).

4. *"Figures 14, 18, 20, 21, 22, and 24 could be normalized to account for the different inflow strengths of the different cases as given in table 4."*

In this work, the focus is on the distribution of the surface pressure fluctuations and not on the amplitude. Therefore, the main conclusions are not affected by normalization with the inflow velocity. In the opinion of the authors, this would only complicate the interpretation of the figures. Furthermore, we consider the use of $p$ instead of $c_p$ more appropriate with regard to the broader objective of assessing noise emissions.

Therefore, the authors would prefer to keep showing the pressure fluctuations and rather than the pressure coefficient fluctuations.

5. *"line 570: "Both mean velocities and turbulence up to 1Hz are realistically captured at the turbine position." –> See discussions above. The difference in TI is 5 to 10 percentage points at the turbine location in the FLOWER simulation. This is significant and requires a more balanced discussion to explain this to the reader."*

As discussed in $\boxed{\textbf{R3:G1-c}}$ (page 16, line 350), the $TI$ values resulting from the simulation cannot really be compared with measurements because of the coarser grid resolution at the mast position. Therefore, these values are removed from the manuscript. The fairest possible comparison is between the simulation one rotor radius in front of the turbine (x=-R, y=0, z=77 m a.g.l.) and the met mast 20 at 80 m a.g.l..

6. *"line 573: "The characteristics of the turbine wake can be compared with lidar measurements, for example, and are well represented in the simulation." –> Where is this comparison of the measured and simulated wake presented (sorry in case I missed this)."*

Section 5.1 (page 17) describes the comparison made with respect to the wake. Figure 10 (page 18) shows the velocity deficit caused by the wake in both the simulation and the measurement. Isolines are added to improve the visibility of the wake.

7. *"Several figure captions: D from mast 20 –> As "D" is typically used for turbine diameter, it is not ideal to use its abbreviation for "distance.""*

The variable $D$ is removed from the manuscript and "distance" is written out in the titles of the x-axes in Figure 7 (page 15), Figure 8 (page 16) and Figure 10 (page 18).

8. *"Figure 9 caption "WS" in between brackets"*

$WS$ is used as a variable for wind speed and not as an abbreviation. To avoid confusion $WS$ is removed from the caption of Figure 9 (page 17).

9. "*Figure 10a at the top ==> should this be with and without turbine instead of "terrain - empty"?*"

The heading of Figure 10 (page 18)(a) is adapted to the new case names. This makes it clearer that the difference between the simulation with and without turbine is meant.

10. "*3.2 "Unsteady terrain simulation" ==> The terrain itself is not unsteady.*"

Adjusted the title of Section 3.2 (page 6) and Section 4.2 (page 15), see R3:S10-a (page 6, line 158) and R3:S10-b (page 15, line 335).

11. "*line 158: "following the literature mentioned in Sect. 1.1." –> can these references be specified? Many studies are cited in 1.1*"

The references are added in the manuscript, see R3:S11-a (page 6, line 160).

12. "*line 166: "a pre-run with an increased time step (70 t) is utilized." –> Can you give a reference to this procedure? It would seem that this makes your timestep too large to satisfy CFL conditions.*"

This is a best practice at the authors' institute for simulations with a prescribed velocity profile. As this is only relevant for the deployed implementation of the boundary condition in *FLOWer*, it is removed from the manuscript. Instead, information about the physical initialisation is added, see R3:S12-a (page 7, line 169).

This time step indeed leads to CFL numbers greater than one. However, this pre-run does not serve to develop a physical flow field in the domain, but only to propagate or dissipate pressure and density disturbances from the CFD domain that arise from inconsistent initialisation of variables at the Dirichlet-BC. The physical initialisation is then done by propagating the resolved turbulence through the domain with a time step that satisfies the CFL condition (see Section 3.2 (page 6)), as now stated.

13. "*line 185: "BL cells with reduced resolution". The use of "reduced resolution is confusing; use, for example ", cells with smaller vertical extent."*"

Adjusted as suggested, see R3:S13-a (page 8, line 189).

14. "*Double-check for spelling/grammar*"

The authors have revised the spelling and grammar, see among others R3:S14-a (page 5, line 121) to R3:S14-k (page 30, line 614). In addition, in the case of acceptance, the final revised paper is copy-edited for English, typeset, and proofread by copernicus before publication.

[revised manuscript text omitted]